# CFG++: Manifold-constrained Classifier Free Guidance for Diffusion Models

**Hyungjin Chung**[*]**, Jeongsol Kim**[*]**, Geon Yeong Park**[*]**, Hyelin Nam**[*]**, Jong Chul Ye**
KAIST
[*]: Equal Contribution
{hj.chung, jeongsol, pky3436, hyelin.nam, jong.ye}@kaist.ac.kr

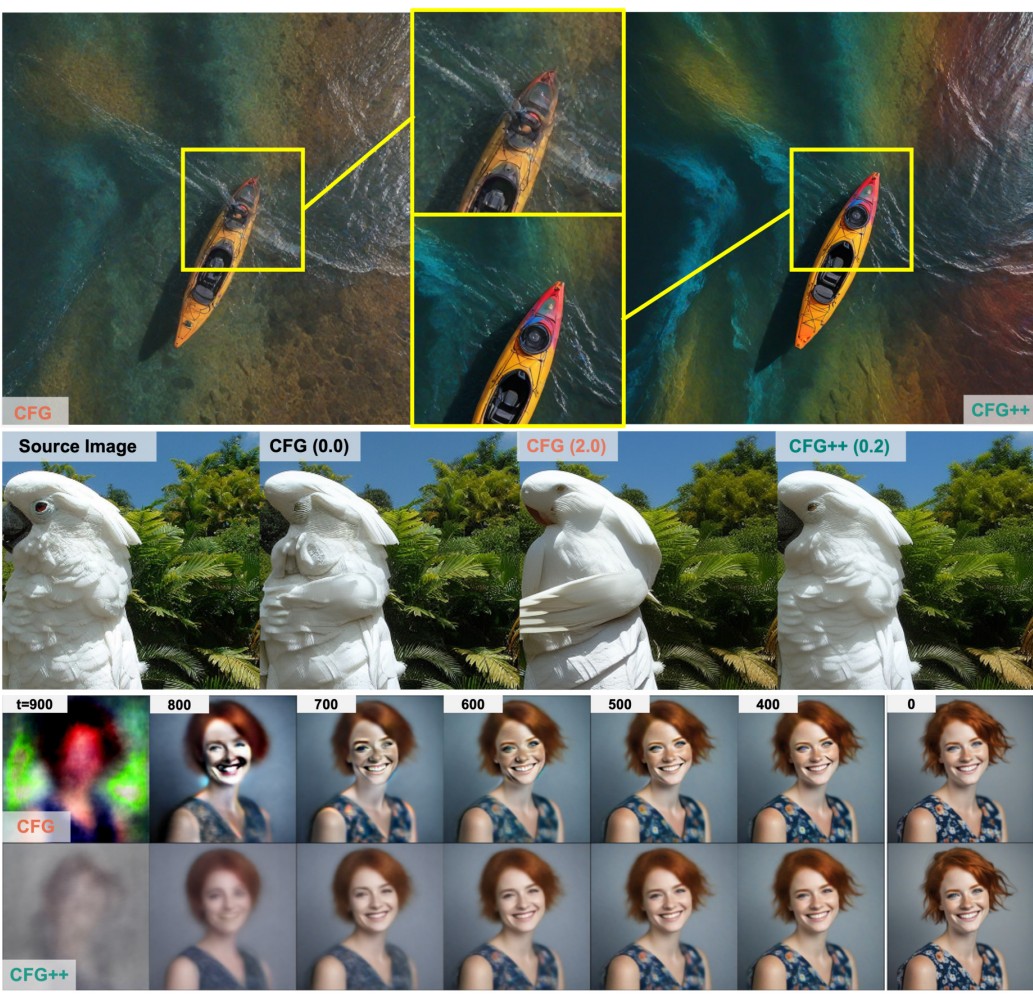

Figure 1: **(Top)** Comparison of T2I results by SDXL-Turbo for the prompt "kayak in the water, optical color, aerial view, rainbow". The CFG-guided image has significant artifacts, which are reduced in the CFG++ version. **(Middle)** DDIM Inversion results under CFG show noticeable artifacts at various CFG scales, which are significantly reduced by CFG++. **(Bottom)** The evolution of denoised estimates differs between CFG and CFG++. CFG exhibits sudden shifts and intense color saturation early in reverse diffusion, while CFG++ transitions smoothly from low to high-resolution.

## Abstract

Classifier-free guidance (CFG) is a fundamental tool in modern diffusion models for text-guided generation. Although effective, CFG has notable drawbacks. For instance, DDIM with CFG lacks invertibility, complicating image editing; furthermore, high guidance scales, essential for high-quality outputs, frequently

result in issues like mode collapse. This paper reveals that the problems may stem from the off-manifold phenomenon associated with CFG, rather than the diffusion models themselves. More specifically, inspired by the recent advancements of diffusion model-based inverse problem solvers (DIS), we reformulate text-guidance as an inverse problem with a text-conditioned score matching loss and develop CFG++, a novel approach that tackles the off-manifold challenges inherent in traditional CFG. CFG++ features a surprisingly simple fix to CFG, yet it offers significant improvements, including better sample quality for text-to-image generation, invertibility, smaller guidance scales, etc. Furthermore, CFG++ enables seamless interpolation between unconditional and conditional sampling at lower guidance scales, consistently outperforming traditional CFG at all scales. Moreover, CFG++ can be easily integrated into high-order diffusion solvers and naturally extends to distilled diffusion models. Experimental results confirm that our method significantly enhances performance in text-to-image generation, DDIM inversion, editing, and solving inverse problems, suggesting a wide-ranging impact and potential applications in various fields that utilize text guidance. Project Page: https://cfgpp-diffusion.github.io.

# 1 INTRODUCTION

Classifier-free guidance (CFG) (Ho & Salimans, 2021) forms the key basis of modern text-guided generation with diffusion models (Dhariwal & Nichol, 2021; Rombach et al., 2022). Nowadays, it is common practice to train a diffusion model with large-scale paired text-image data (Schuhmann et al., 2022), so that sampling (i.e. generating) a signal (e.g. image, video) from a diffusion model can either be done unconditionally from $p_\theta(\boldsymbol{x}|\varnothing) \equiv p_\theta(\boldsymbol{x})$, or conditionally from $p_\theta(\boldsymbol{x}|\boldsymbol{c})$, where $\boldsymbol{c}$ is the text conditioning. Once trained, it seems natural that one would acquire samples from the conditional distribution by simply solving the probability-flow ODE or SDE sampling (Song et al., 2021b;a; Karras et al., 2022) with the conditional score function. In practice, however, it is observed that the conditioning signal is insufficient when used naively. To emphasize the guidance, one uses the guidance scale $\omega > 1$, where the direction can be defined by the direction from the unconditional score to the conditional score (Ho & Salimans, 2021).

In modern text-to-image (T2I) diffusion models, the guidance scale $\omega$ is typically set within the range of $[5.0, 30]$, referred to as the *moderately* high range of CFG guidance (Chen et al., 2024; Podell et al., 2023). The insufficiency in guidance also holds for classifier guidance (Dhariwal & Nichol, 2021; Song et al., 2021b) so that a scale of 10 was used. While using a high guidance scale yields higher-quality images with better alignment to the condition, it is also prone to mode collapse, reduces sample diversity, and yields an inevitable accumulation of errors during the sampling process. One example is DDIM inversion (Dhariwal & Nichol, 2021), a pivotal technique for controllable synthesis and editing (Mokady et al., 2023), where running the inversion process with $\omega > 1.0$ leads to significant compromise in the reconstruction performance (Mokady et al., 2023; Wallace et al., 2023). Another extreme example would be score distillation sampling (SDS) (Poole et al., 2022), where the guidance scale in the order of a few hundred is chosen. Using such a high guidance scale leads to better asset quality to some extent, but induces blurry and saturated results. Several research efforts have been made to mitigate this downside by exploring methods where using a smaller guidance scale suffices (Wang et al., 2024; Liang et al., 2023). Although recent progress in SDS-type methods has reduced the necessary guidance scale to a range that is similar to those of ancestral samplers, using a moderately large $\omega$ is considered an inevitable choice.

In this work, we aim to give an answer to this conundrum by revisiting the geometric view of diffusion models. In particular, inspired by the recent advances in diffusion-based inverse problem solvers (DIS) (Kadkhodaie & Simoncelli, 2021; Chung et al., 2023a; Song et al., 2023; Kim et al., 2024b; Chung et al., 2024), we reformulate the text guidance as an inverse problem with a text-conditioned score-matching loss and derive a reverse diffusion sampling strategy by utilizing decomposed diffusion sampling (DDS) (Chung et al., 2024). This results in a surprisingly simple fix of CFG to the sampling process without any computational overhead. The resulting process, which we call CFG++, works with a small guidance scale, typically $\lambda \in [0.0, 1.0]$, that smoothly *interpolates* between unconditional and conditional sampling, with $\lambda = 1.0$ having a similar effect as using CFG sampling with $\omega \sim 12.5$ at 50 neural function evaluation (NFE). Furthermore, CFG++ reduces the inversion

---

**Algorithm 1** Reverse Diffusion with CFG

**Require:** $\boldsymbol{x}_T \sim \mathcal{N}(0, \mathbf{I}_d), 0 \leq \omega \in \mathbb{R}$
1: **for** $i = T$ **to** 1 **do**
2: $\quad \hat{\boldsymbol{\epsilon}}_c^\omega(\boldsymbol{x}_t) = \hat{\boldsymbol{\epsilon}}_\varnothing(\boldsymbol{x}_t) + \omega[\hat{\boldsymbol{\epsilon}}_c(\boldsymbol{x}_t) - \hat{\boldsymbol{\epsilon}}_\varnothing(\boldsymbol{x}_t)]$
3: $\quad \hat{\boldsymbol{x}}_c^\omega(\boldsymbol{x}_t) \leftarrow (\boldsymbol{x}_t - \sqrt{1 - \bar{\alpha}_t}\hat{\boldsymbol{\epsilon}}_c^\omega(\boldsymbol{x}_t))/\sqrt{\bar{\alpha}_t}$
4: $\quad \boldsymbol{x}_{t-1} = \sqrt{\bar{\alpha}_{t-1}}\hat{\boldsymbol{x}}_c^\omega(\boldsymbol{x}_t) + \sqrt{1 - \bar{\alpha}_{t-1}}\hat{\boldsymbol{\epsilon}}_c^\omega(\boldsymbol{x}_t)$
5: **end for**
6: **return** $\boldsymbol{x}_0$

**Algorithm 2** Reverse Diffusion with CFG++

**Require:** $\boldsymbol{x}_T \sim \mathcal{N}(0, \mathbf{I}_d), \lambda \in [0, 1]$
1: **for** $i = T$ **to** 1 **do**
2: $\quad \hat{\boldsymbol{\epsilon}}_c^\lambda(\boldsymbol{x}_t) = \hat{\boldsymbol{\epsilon}}_\varnothing(\boldsymbol{x}_t) + \lambda[\hat{\boldsymbol{\epsilon}}_c(\boldsymbol{x}_t) - \hat{\boldsymbol{\epsilon}}_\varnothing(\boldsymbol{x}_t)]$
3: $\quad \hat{\boldsymbol{x}}_c^\lambda(\boldsymbol{x}_t) \leftarrow (\boldsymbol{x}_t - \sqrt{1 - \bar{\alpha}_t}\hat{\boldsymbol{\epsilon}}_c^\lambda(\boldsymbol{x}_t))/\sqrt{\bar{\alpha}_t}$
4: $\quad \boldsymbol{x}_{t-1} = \sqrt{\bar{\alpha}_{t-1}}\hat{\boldsymbol{x}}_c^\lambda(\boldsymbol{x}_t) + \sqrt{1 - \bar{\alpha}_{t-1}}\hat{\boldsymbol{\epsilon}}_\varnothing(\boldsymbol{x}_t)$
5: **end for**
6: **return** $\boldsymbol{x}_0$

---

Figure 2: Comparison between reverse diffusion process by CFG and CFG++. CFG++ proposes a simple but surprisingly effective fix: using $\hat{\boldsymbol{\epsilon}}_\varnothing(\boldsymbol{x}_t)$ instead of $\hat{\boldsymbol{\epsilon}}_c^\omega(\boldsymbol{x}_t)$ in updating $\boldsymbol{x}_{t-1}$.

error, enhancing and simplifying image reconstruction, as well as editing. Comparing CFG++ against CFG shows that we achieve consistently better sample quality for text-to-image (T2I) generation, significantly better DDIM inversion capabilities that lead to enhanced reconstruction and editing, and enabling the incorporation of CFG guidance to diffusion inverse solvers (DIS) (Chung et al., 2023a). While the applications of CFG++ that we show in this work are limited, we believe that our work will have a broad impact that can be applied to all applications that leverage text guidance through the traditional CFG.

## 2 BACKGROUND

**Diffusion models.** Diffusion models (Ho et al., 2020; Song et al., 2021b; Karras et al., 2022) are generative models designed to learn the reversal of a forward noising process. This process starts with an initial distribution $p_0(\boldsymbol{x})$ where $\boldsymbol{x} \in \mathbb{R}^n$, and progresses towards the standard Gaussian distribution $p_T(\boldsymbol{x}) \approx \mathcal{N}(\boldsymbol{0}, \boldsymbol{I})$, utilizing forward Gaussian perturbation kernels. Sampling from the data distribution can be performed by solving either the reverse stochastic differential equation (SDE) or the equivalent probability-flow ordinary differential equation (PF-ODE) (Song et al., 2021b). For example, under the choice $p(\boldsymbol{x}_t|\boldsymbol{x}_0) = \mathcal{N}(\boldsymbol{x}_0, \sigma_t^2\boldsymbol{I})$, the generative PF-ODE reads

$$d\boldsymbol{x}_t = -\sigma_t \nabla_{\boldsymbol{x}_t} \log p(\boldsymbol{x}_t)\, dt = \frac{\boldsymbol{x}_t - \mathbb{E}[\boldsymbol{x}_0|\boldsymbol{x}_t]}{\sigma_t}\, dt, \quad \boldsymbol{x}_T \sim p_T(\boldsymbol{x}_T), \tag{1}$$

where Tweedie's formula (Efron, 2011) $\mathbb{E}[\boldsymbol{x}_0|\boldsymbol{x}_t] = \boldsymbol{x}_t + \sigma_t^2 \nabla_{\boldsymbol{x}_t} \log p(\boldsymbol{x}_t)$ is applied to achieve the second equality. When aiming for a text-conditional diffusion model that can condition on arbitrary $\boldsymbol{c}$, one can extend epsilon matching to a conditional one where the condition is dropped with a certain probability (Ho & Salimans, 2021) so that null conditioning with $\boldsymbol{c} = \varnothing$ is possible. The neural network architecture of $\boldsymbol{\epsilon}_\theta$ is designed so that the condition $\boldsymbol{c}$ can *modulate* the output through cross attention (Rombach et al., 2022). For simplicity in notation throughout the paper, we define $\hat{\boldsymbol{\epsilon}}_c := \boldsymbol{\epsilon}_\theta(\boldsymbol{x}_t, \boldsymbol{c})$ and $\hat{\boldsymbol{\epsilon}}_\varnothing := \boldsymbol{\epsilon}_\theta(\boldsymbol{x}_t, \varnothing)$ by dropping $\theta$ and $\boldsymbol{x}_t$.

Extending the result of Tweedie's formula to the unconditional case under the variance preserving (VP) framework of DDPMs (Ho et al., 2020), we have

$$\mathbb{E}[\boldsymbol{x}_0|\boldsymbol{x}_t, \varnothing] = \hat{\boldsymbol{x}}_\varnothing(\boldsymbol{x}_t) := (\boldsymbol{x}_t - \sqrt{1 - \bar{\alpha}_t}\hat{\boldsymbol{\epsilon}}_\varnothing(\boldsymbol{x}_t))/\sqrt{\bar{\alpha}_t}. \tag{2}$$

Leveraging (2), it is common to use DDIM sampling (Song et al., 2021b) to solve the conditional probability-flow ODE (PF-ODE) of the generative process. Specifically, a single iteration reads

$$\hat{\boldsymbol{x}}_\varnothing = (\boldsymbol{x}_t - \sqrt{1 - \bar{\alpha}_t}\hat{\boldsymbol{\epsilon}}_\varnothing)/\sqrt{\bar{\alpha}_t} \tag{3}$$

$$\boldsymbol{x}_{t-1} = \sqrt{\bar{\alpha}_{t-1}}\hat{\boldsymbol{x}}_\varnothing + \sqrt{1 - \bar{\alpha}_{t-1}}\hat{\boldsymbol{\epsilon}}_\varnothing, \tag{4}$$

where $\hat{\boldsymbol{x}}_\varnothing := \hat{\boldsymbol{x}}_\varnothing(\boldsymbol{x}_t) = \mathbb{E}[\boldsymbol{x}_0|\boldsymbol{x}_t, \varnothing]$ is the denoised signal by Tweedie's formula, and (4) corresponds to the *renoising* step[1]. This is repeated for $t = T, T-1, \ldots, 1$.

For modern diffusion models, it is common to train a diffusion model in the latent space (Rombach et al., 2022) with the latent variable $\boldsymbol{z}$. While most of our experiments are performed with latent diffusion models (LDM), as our framework holds both for pixel- and latent-diffusion models, we will simply use the notation $\boldsymbol{x}$ regardless of the choice.

**Classifier free guidance.** For a conditional diffusion, Ho & Salimans (2021) considered the sharpened posterior distribution $p^\omega(\boldsymbol{x}|\boldsymbol{c}) \propto p(\boldsymbol{x})p(\boldsymbol{c}|\boldsymbol{x})^\omega$. Using Bayes rule for some timestep $t$,

$$\nabla_{\boldsymbol{x}} \log p^\omega(\boldsymbol{x}_t|\boldsymbol{c}) = \nabla_{\boldsymbol{x}_t} \log p(\boldsymbol{x}_t) + \omega(\nabla_{\boldsymbol{x}_t} \log p(\boldsymbol{x}_t|\boldsymbol{c}) - \nabla_{\boldsymbol{x}_t} \log p(\boldsymbol{x}_t)) \tag{5}$$

---

[1]Similarly, we define $\hat{\boldsymbol{x}}_c := \hat{\boldsymbol{x}}_c(\boldsymbol{x}_t) = \mathbb{E}[\boldsymbol{x}_0|\boldsymbol{x}_t, \boldsymbol{c}]$

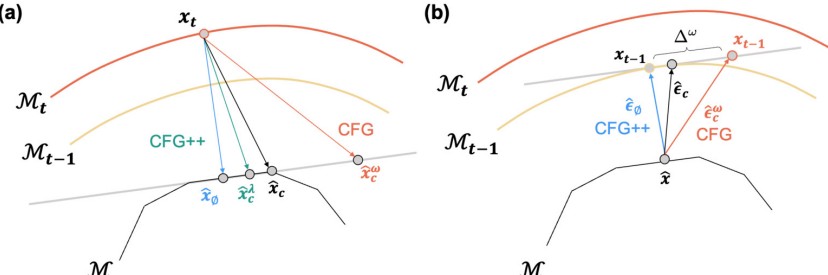

Figure 3: Off-manifold phenomenon of CFG arise from: (a) the typical CFG scale $\omega > 1.0$ which leads to extrapolation and deviation from the piecewise linear data manifold, and (b) CFG's renoising process, which introduces a nonzero offset $\Delta^\omega$ from the correct manifold. CFG++ effectively mitigates all these artifacts.

Parametrizing the score function with $\epsilon_\theta$, as in DDPM (Ho et al., 2020), we have

$$\hat{\epsilon}_c^\omega(\boldsymbol{x}_t) := \hat{\epsilon}_\varnothing(\boldsymbol{x}_t) + \omega[\hat{\epsilon}_c(\boldsymbol{x}_t) - \hat{\epsilon}_\varnothing(\boldsymbol{x}_t)] \tag{6}$$

where we introduce a compact notation $\hat{\epsilon}_c^\omega$ that guides the sampling from the sharpened posterior. When sampling with CFG guidance with DDIM sampling, one replaces $\hat{\epsilon}_\varnothing$ with $\hat{\epsilon}_c^\omega$ for both the Tweedie estimate (3) and the subsequent update step (4), leading to Algorithm 1.

**Diffusion model-based inverse problem solvers.** Diffusion model-based inverse problem solvers (DIS) aims to perform posterior sampling from an unconditional diffusion model (Kadkhodaie & Simoncelli, 2021; Chung et al., 2023a; Song et al., 2023; Kim et al., 2024b). Specifically, for a given loss function $\ell(\boldsymbol{x})$ which often stems from the likelihood, the goal of DIS is to address the optimization problem $\min_{\boldsymbol{x}\in\mathcal{M}} \ell(\boldsymbol{x})$, where $\mathcal{M}$ represents the clean data manifold sampled from the unconditional distribution $p_0(\boldsymbol{x})$. It is essential to navigate in a way that minimizes cost while also identifying the correct clean manifold.

Chung et al. (2023a) proposed diffusion posterior sampling (DPS), where the updated estimate from the noisy sample $\boldsymbol{x}_t \in \mathcal{M}_t$ is constrained to stay on the same noisy manifold $\mathcal{M}_t$. This is achieved by computing the manifold constrained gradient (MCG) (Chung et al., 2022) on a noisy sample $\boldsymbol{x}_t \in \mathcal{M}_t$ as $\nabla_{\boldsymbol{x}_t}^{mcg}\ell(\boldsymbol{x}_t) := \nabla_{\boldsymbol{x}_t}\ell(\hat{\boldsymbol{x}}_t)$, where $\hat{\boldsymbol{x}}_t$ is the denoised sample in (2) through Tweedie's formula (Efron, 2011). The resulting algorithm with DDIM (Song et al., 2021b) can be stated as follows:

$$\boldsymbol{x}_{t-1} = \sqrt{\bar{\alpha}_{t-1}}\left(\hat{\boldsymbol{x}}_\varnothing - \gamma_t \nabla_{\boldsymbol{x}_t}\ell(\hat{\boldsymbol{x}}_\varnothing)\right) + \sqrt{1-\bar{\alpha}_{t-1}}\hat{\epsilon}_\varnothing, \tag{7}$$

where $\gamma_t > 0$ denotes the step size. Under the linear manifold assumption (Chung et al., 2022; 2023a), this allows precise transition to $\mathcal{M}_{t-1}$. To mitigate the computational complexity and the instability of neural network backprop, Chung et al. (2024) shows that (7) can be equivalently represented as

$$\boldsymbol{x}_{t-1} \simeq \sqrt{\bar{\alpha}_{t-1}}\left(\hat{\boldsymbol{x}}_\varnothing - \gamma_t \nabla_{\hat{\boldsymbol{x}}_\varnothing}\ell(\hat{\boldsymbol{x}}_\varnothing)\right) + \sqrt{1-\bar{\alpha}_{t-1}}\hat{\epsilon}_\varnothing \tag{8}$$

under further assumptions on $\mathcal{M}$. This method, often referred to as the decomposed diffusion sampling (DDS), bypasses the computation of the score Jacobian, similar to Poole et al. (2022), making it stable and suitable for large-scale medical imaging inverse problems (Chung et al., 2024). In the following, we leverage the insight from DDS to propose an algorithm to improve upon the CFG algorithm.

## 3 CFG++ : MANIFOLD-CONSTRAINED CFG

### 3.1 DERIVATION OF CFG++

Instead of uncritically adopting the sharpened posterior distribution $p^\omega(\boldsymbol{x}|\boldsymbol{c}) \propto p(\boldsymbol{x})p(\boldsymbol{c}|\boldsymbol{x})^\omega$ as introduced by Ho & Salimans (2021), we adopt a fundamentally different strategy by reformulating text-guidance as an optimization problem. Specifically, our focus is on identifying a loss function $\ell(\boldsymbol{x})$ in (8) such that, when minimized under the condition set by the text, enables the reverse diffusion process to generate samples that increasingly satisfy the text condition progressively.

One of the most significant contributions of this paper is to reveal that the text-conditioned score matching loss or alternatively, score distillation sampling (SDS) loss (Poole et al., 2022) is ideally

suited for our purpose. Specifically, we are interested in solving the following inverse problem through diffusion models:

$$\min_{\boldsymbol{x}\in\mathcal{M}} \ell_{sds}(x), \quad \ell_{sds}(\boldsymbol{x}) := \|\boldsymbol{\epsilon}_\theta(\sqrt{\bar{\alpha}_t}\boldsymbol{x} + \sqrt{1-\bar{\alpha}_t}\boldsymbol{\epsilon}, \boldsymbol{c}) - \boldsymbol{\epsilon}\|_2^2 \tag{9}$$

This implies that our goal is to identify solutions on the clean manifold $\mathcal{M}$ that optimally aligns with the text condition $\boldsymbol{c}$.

To avoid the Jacobian computation, in this paper, we attempt to solve (9) through DDS in (8). The resulting sampling process from reverse diffusion is then given by

$$\boldsymbol{x}_{t-1} \quad = \quad \sqrt{\bar{\alpha}_{t-1}}\left(\hat{\boldsymbol{x}}_\varnothing - \gamma_t \nabla_{\hat{\boldsymbol{x}}_\varnothing}\ell_{sds}(\hat{\boldsymbol{x}}_\varnothing)\right) + \sqrt{1-\bar{\alpha}_{t-1}}\hat{\boldsymbol{\epsilon}}_\varnothing. \tag{10}$$

By using $\boldsymbol{x}_t = \sqrt{\bar{\alpha}_t}\boldsymbol{x} + \sqrt{1-\bar{\alpha}_t}\boldsymbol{\epsilon}$ from the clean image $\boldsymbol{x} \in \mathcal{M}$, we can equivalently write the loss as $\ell_{sds}(\boldsymbol{x}) = \frac{\bar{\alpha}_t}{1-\bar{\alpha}_t}\|\boldsymbol{x} - \hat{\boldsymbol{x}}_{\boldsymbol{c}}\|^2$, which leads to

$$\boldsymbol{x}_{t-1} = \sqrt{\bar{\alpha}_{t-1}}\left(\hat{\boldsymbol{x}}_\varnothing + \lambda(\hat{\boldsymbol{x}}_{\boldsymbol{c}} - \hat{\boldsymbol{x}}_\varnothing)\right) + \sqrt{1-\bar{\alpha}_{t-1}}\hat{\boldsymbol{\epsilon}}_\varnothing \tag{11}$$

where $\lambda := \frac{2\bar{\alpha}_t}{1-\bar{\alpha}_t}\gamma_t$. Using the CFG notation $\hat{\boldsymbol{\epsilon}}_{\boldsymbol{c}}^\lambda(\boldsymbol{x}_t) := \hat{\boldsymbol{\epsilon}}_\varnothing(\boldsymbol{x}_t) + \lambda[\hat{\boldsymbol{\epsilon}}_{\boldsymbol{c}}(\boldsymbol{x}_t) - \hat{\boldsymbol{\epsilon}}_\varnothing(\boldsymbol{x}_t)]$ and $\hat{\boldsymbol{x}}_\varnothing + \lambda(\hat{\boldsymbol{x}}_{\boldsymbol{c}} - \hat{\boldsymbol{x}}_\varnothing) = (\boldsymbol{x}_t - \sqrt{1-\bar{\alpha}_t}\hat{\boldsymbol{\epsilon}}_{\boldsymbol{c}}^\lambda(\boldsymbol{x}_t))/\sqrt{\bar{\alpha}_t}$, (11) can be equivalently represented as

$$\hat{\boldsymbol{x}}_{\boldsymbol{c}}^\lambda(\boldsymbol{x}_t) = (\boldsymbol{x}_t - \sqrt{1-\bar{\alpha}_t}\hat{\boldsymbol{\epsilon}}_{\boldsymbol{c}}^\lambda(\boldsymbol{x}_t))/\sqrt{\bar{\alpha}_t} \tag{12}$$

$$\boldsymbol{x}_{t-1} = \sqrt{\bar{\alpha}_{t-1}}\hat{\boldsymbol{x}}_{\boldsymbol{c}}^\lambda(\boldsymbol{x}_t) + \sqrt{1-\bar{\alpha}_{t-1}}\hat{\boldsymbol{\epsilon}}_\varnothing(\boldsymbol{x}_t) \tag{13}$$

which is summarized in Algorithm 2. By examining Algorithm 1 and Algorithm 2, we observe that CFG and CFG++ are mostly the same, with a crucial difference in the renoising process. This surprisingly simple fix of utilizing the unconditional noise $\hat{\boldsymbol{\epsilon}}_\varnothing(\boldsymbol{x}_t)$ instead of $\hat{\boldsymbol{\epsilon}}_{\boldsymbol{c}}^\omega(\boldsymbol{x}_t)$ leads to a smoother trajectory of generation, (Fig. 1 bottom) and generation with superior quality (Fig. 1 top).

Although we focus our construction of the solver on DDIM for simplicity, note that DDIM is just one way of reverse sampling. There are other widely-used solvers such as Karras Euler and its variants (Karras et al., 2022), DPM-solver (Lu et al., 2022a;b), their ancestral variants[2], etc. For most widely used solvers up to the second order, a single-step update of solving the unconditional PF-ODE can be represented as

$$\boldsymbol{x}_i = \hat{\boldsymbol{x}}_\varnothing(\boldsymbol{x}_{i-1}) + a_i\hat{\boldsymbol{x}}_\varnothing(\boldsymbol{x}_{i-1}) + b_i\hat{\boldsymbol{x}}_\varnothing(\boldsymbol{x}_{i-2}) + c_i\boldsymbol{x}_{i-1} + d_i\boldsymbol{\epsilon}, \ \boldsymbol{\epsilon} \sim \mathcal{N}(0, \boldsymbol{I}), \tag{14}$$

with $d_i \neq 0$ when one uses an ancestral sampler. Note that the first right-hand side term in (14) corresponds to the denosing, whereas the rest terms describes the higher-order corrected version of the *renoising* process. As the goal of CFG++ is to optimize the denoising process under the text-guidance while keeping the renoising components equivalent to unconditional sampling, applying CFG++ to the general iteration in (14) simply leads to

$$\boldsymbol{x}_i = \left(\hat{\boldsymbol{x}}_\varnothing(\boldsymbol{x}_{i-1}) - \lambda\nabla_{\hat{\boldsymbol{x}}_\varnothing(\boldsymbol{x}_{i-1})}\ell_{sds}(\hat{\boldsymbol{x}}_\varnothing(\boldsymbol{x}_{i-1}))\right) + a_i\hat{\boldsymbol{x}}_\varnothing(\boldsymbol{x}_{i-1}) + b_i\hat{\boldsymbol{x}}_\varnothing(\boldsymbol{x}_{i-2}) + c_i\boldsymbol{x}_{i-1} + d_i\boldsymbol{\epsilon}$$

$$= \hat{\boldsymbol{x}}_{\boldsymbol{c}}^\lambda(\boldsymbol{x}_{i-1}) + a_i\hat{\boldsymbol{x}}_\varnothing(\boldsymbol{x}_{i-1}) + b_i\hat{\boldsymbol{x}}_\varnothing(\boldsymbol{x}_{i-2}) + c_i\boldsymbol{x}_{i-1} + d_i\boldsymbol{\epsilon}, \ \boldsymbol{\epsilon} \sim \mathcal{N}(0, \boldsymbol{I}). \tag{15}$$

Moreover, CFG++ naturally extends to distilled diffusion models, such as SDXL-turbo (Sauer et al., 2023) and SDXL-lightning (Lin et al., 2024), where akin to (15), we use the conditional denoised estimate, but for the rest of the noise components in the Euler solver, we use the unconditional estimate. For details in how to apply CFG++ to various solvers, see Appendix A.

## 3.2 GEOMETRY OF CFG++

**Mitigating off-manifold phenomenon.** In Fig. 1 (bottom), we illustrate the evolution of the posterior mean through Tweedie's formula during the reverse diffusion process. Notably, in the early phases of reverse diffusion sampling under CFG, there is a sudden shift in the image and intense color saturation. Conversely, CFG++ is free from the undesirable off-manifold phenomenon. In the following, we investigate why this is the case.

Note that the denoised estimate of CFG and CFG++ at time $t$ can be equivalently represented as

$$\hat{\boldsymbol{x}}_{\boldsymbol{c}}^\lambda(\boldsymbol{x}_t) = (1-\lambda)\hat{\boldsymbol{x}}_\varnothing(\boldsymbol{x}_t) + \lambda\hat{\boldsymbol{x}}_{\boldsymbol{c}}(\boldsymbol{x}_t), \quad \hat{\boldsymbol{x}}_{\boldsymbol{c}}^\omega(\boldsymbol{x}_t) = (1-\omega)\hat{\boldsymbol{x}}_\varnothing(\boldsymbol{x}_t) + \omega\hat{\boldsymbol{x}}_{\boldsymbol{c}}(\boldsymbol{x}_t). \tag{16}$$

---

[2]https://github.com/crowsonkb/k-diffusion

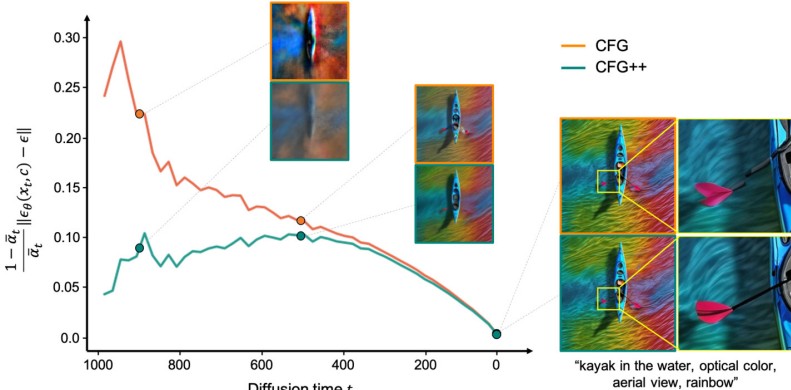

Figure 4: Text-conditioned score matching loss throughout the reverse diffusion sampling for both CFG and CFG++ in SDXL. Avg. loss computed with 55 prompts from (Chen et al., 2024).

$\lambda, \omega \in [0, 1]$ facilitates an *interpolation*. However, with $\omega > 1.0$, CFG *extrapolates* beyond the unconditional and conditional estimates. Consequently, under the assumption that the clean manifold be piecewise linear (Chung et al., 2024), the conditional posterior mean estimates from CFG obtained with a guidance scale outside the range of $[0, 1]$, can easily extend beyond the piecewise linear manifold. This may lead to the estimates potentially "falling off" the data manifold, as depicted by an orange arrow pointing downwards in Fig. 3(a). Thus, we select $\lambda \in [0, 1]$ as the guidance scale for CFG++ to ensure it remains an *interpolation* between the unconditional and conditional estimate, thus preventing it from 'falling off' the clean data manifold. An additional source of the off-manifold phenomenon in CFG occurs during the transition from the clean manifold $\mathcal{M}$ to the subsequent noisy manifold $\mathcal{M}_{t-1}$, due to a similar off-manifold phenomenon from the large guidance scale (i.e. extrapolation), illustrated in Fig. 3 (b).

**Text-Image alignment.** In Fig. 1 (top), and Fig. 10, we observe enhanced text-to-image alignment achieved with CFG++. This enhanced alignment capability is a natural consequence of CFG++, which directly minimizes the text-conditioned score-matching loss as shown in (9). In contrast, CFG indirectly seeks text alignment through the sharpened posterior distribution $p^{\omega}(\boldsymbol{x}|\boldsymbol{c}) \propto p(\boldsymbol{x})p(\boldsymbol{c}|\boldsymbol{x})^{\omega}$. Therefore, CFG++ inherently outperforms CFG in terms of text alignment due to its fundamental design principle. For example, Fig. 4 displays the normalized text-conditioned score matching loss, represented as $(1 - \bar{\alpha}_t)\|\boldsymbol{\epsilon} - \boldsymbol{\epsilon}_\theta(\boldsymbol{x}_t, \boldsymbol{c})\|^2/\bar{\alpha}_t = \|\boldsymbol{x} - \hat{\boldsymbol{x}}_c\|^2$, throughout the reverse diffusion sampling process for both CFG and CFG++. The loss plot associated with CFG shows fluctuations and maintains a noticeable gap compared to CFG++ even after the completion of the reverse diffusion process. In fact, the fluctuation associated with CFG is also related to the off-manifold issue, and from Fig. 4 we can easily see that the off-manifold phenomenon is more dominant at early stage of reverse diffusion sampling. Conversely, the loss trajectory for CFG++ demonstrates a much smoother variation, particularly during the early stages of reverse diffusion.

**DDIM inversion.** As discussed in (Song et al., 2021a), the denoising process for unconditional DDIM is approximately invertible, meaning that $\boldsymbol{x}_t$ can generally be recovered from $\boldsymbol{x}_{t-1}$ Specifically, from (3) and (4), we have the following approximate inversion formula for unconditional DDIM:

$$\hat{\boldsymbol{x}}_{\varnothing}(\boldsymbol{x}_t) = (\boldsymbol{x}_{t-1} - \sqrt{1 - \bar{\alpha}_{t-1}}\hat{\boldsymbol{\epsilon}}_{\varnothing}(\boldsymbol{x}_t))/\sqrt{\bar{\alpha}_{t-1}} \simeq (\boldsymbol{x}_{t-1} - \sqrt{1 - \bar{\alpha}_{t-1}}\hat{\boldsymbol{\epsilon}}_{\varnothing}(\boldsymbol{x}_{t-1}))/\sqrt{\bar{\alpha}_{t-1}} \quad (17)$$

$$\boldsymbol{x}_t = \sqrt{\bar{\alpha}_t}\hat{\boldsymbol{x}}_{\varnothing}(\boldsymbol{x}_t) + \sqrt{1 - \bar{\alpha}_t}\hat{\boldsymbol{\epsilon}}_{\varnothing}(\boldsymbol{x}_t) \simeq \sqrt{\bar{\alpha}_t}\hat{\boldsymbol{x}}_{\varnothing}(\boldsymbol{x}_t) + \sqrt{1 - \bar{\alpha}_t}\hat{\boldsymbol{\epsilon}}_{\varnothing}(\boldsymbol{x}_{t-1}) \quad (18)$$

where the approximation arises from $\hat{\boldsymbol{\epsilon}}_{\varnothing}(\boldsymbol{x}_t) \simeq \hat{\boldsymbol{\epsilon}}_{\varnothing}(\boldsymbol{x}_{t-1})$. A similar inversion procedure has been employed for conditional DDIM inversion under CFG by assuming $\hat{\boldsymbol{\epsilon}}_c^{\omega}(\boldsymbol{x}_t) \simeq \hat{\boldsymbol{\epsilon}}_c^{\omega}(\boldsymbol{x}_{t-1})$, and replacing $\hat{\boldsymbol{\epsilon}}_{\varnothing}$ by $\hat{\boldsymbol{\epsilon}}_c^{\omega}$ as detailed in Algorithm 3.

On the other hand, by examining (12) and (13), we can obtain the following approximate DDIM inversion formula under CFG++:

$$\hat{\boldsymbol{x}}_c^{\lambda}(\boldsymbol{x}_t) \simeq (\boldsymbol{x}_{t-1} - \sqrt{1 - \bar{\alpha}_{t-1}}\hat{\boldsymbol{\epsilon}}_{\varnothing}(\boldsymbol{x}_{t-1}))/\sqrt{\bar{\alpha}_{t-1}} \quad (19)$$

$$\boldsymbol{x}_t \simeq \sqrt{\bar{\alpha}_t}\hat{\boldsymbol{x}}_c^{\lambda}(\boldsymbol{x}_t) + \sqrt{1 - \bar{\alpha}_t}\hat{\boldsymbol{\epsilon}}_c^{\lambda}(\boldsymbol{x}_{t-1}) \quad (20)$$

| Method | $\omega = 2.0, \lambda = 0.2$ | | | $\omega = 5.0, \lambda = 0.4$ | | | $\omega = 7.5, \lambda = 0.6$ | | | $\omega = 9.0, \lambda = 0.8$ | | | $\omega = 12.5, \lambda = 1.0$ | | |
|---|---|---|---|---|---|---|---|---|---|---|---|---|---|---|---|
| | FID↓ | CLIP↑ | IR↑ | FID↓ | CLIP↑ | IR↑ | FID↓ | CLIP↑ | IR↑ | FID↓ | CLIP↑ | IR↑ | FID↓ | CLIP↑ | IR↑ |
| CFG (Ho & Salimans, 2021) | 13.84 | 0.298 | -0.235 | 15.08 | 0.310 | 0.068 | 17.71 | 0.312 | 0.152 | 20.01 | 0.312 | 0.170 | 21.23 | 0.313 | 0.192 |
| CFG++ (ours) | **12.75** | **0.303** | **-0.218** | **14.95** | 0.310 | **0.071** | **17.47** | 0.312 | **0.156** | **19.34** | **0.313** | **0.208** | **20.88** | 0.313 | **0.194** |

Table 1: Quantitative evaluation of 50 NFE DDIM T2I with SD v1.5 on COCO 10k

where we apply the approximation $\hat{\epsilon}_c^\lambda(x_t) \simeq \hat{\epsilon}_c^\lambda(x_{t-1})$ alongside the usual assumption $\hat{\epsilon}_\varnothing(x_t) \simeq \hat{\epsilon}_\varnothing(x_{t-1})$. This formulation underpins the CFG++ guided DDIM inversion algorithm, presented in Algorithm 4.

In practice, for small time step size the error $\hat{\epsilon}_\varnothing(x_t) \simeq \hat{\epsilon}_\varnothing(x_{t-1})$ is relatively small, so the unconditional DDIM inversion by (17) and (18) lead to relatively insignificant errors. Unfortunately, the corresponding inversion from conditional diffusion under CFG is quite distorted as noted in (Mokady et al., 2023; Wallace et al., 2023). In fact, this distortion is originated from the inaccuracy of the approximation $\hat{\epsilon}_c^\omega(x_t) \simeq \hat{\epsilon}_c^\omega(x_{t-1})$. More specifically, even when $\hat{\epsilon}_\varnothing(x_t) \simeq \hat{\epsilon}_\varnothing(x_{t-1})$, the approximation error from the CFG is given by

$$\varepsilon_{cfg} := \hat{\epsilon}_c^\omega(x_t) - \hat{\epsilon}_c^\omega(x_{t-1}) = (\hat{\epsilon}_\varnothing(x_t) - \hat{\epsilon}_\varnothing(x_{t-1})) + \omega(\delta\hat{\epsilon}_c(x_t) - \delta\hat{\epsilon}_c(x_{t-1}))$$
$$\simeq \omega(\delta\hat{\epsilon}_c(x_t) - \delta\hat{\epsilon}_c(x_{t-1})), \tag{21}$$

where $\delta\hat{\epsilon}_c(x_t) := \hat{\epsilon}_c(x_t) - \hat{\epsilon}_\varnothing(x_t)$ denotes the directional component from unconditional to the conditional diffusion. Since the directional component by the text guidance is not negligible, this error becomes significant for high guidance scale $\omega$. Accordingly, the guidance scale must be heavily downweighted in order for inversions on real world images to be stable, thus limiting the strength of edits. To mitigate this issue, the authors in (Mokady et al., 2023; Wallace et al., 2023) developed null text optimization and coupled transform techniques, respectively.

On the other hand, under the usual DDIM assumption $\hat{\epsilon}_\varnothing(x_t) \simeq \hat{\epsilon}_\varnothing(x_{t-1})$, the approximation error of CFG++ mainly arises from (20), which is smaller than that of CFG since we have

$$\|\varepsilon_{cfg++}\| = \lambda\|\delta\hat{\epsilon}_c(x_t) - \delta\hat{\epsilon}_c(x_{t-1})\| < \|\varepsilon_{cfg}\| \tag{22}$$

thanks to $\lambda < \omega$. Therefore, CFG++ significantly improves the DDIM inversion as shown Fig. 1 (middle) for representative results and Sec. 4.2 for further discussions.

## 4 EXPERIMENTAL RESULTS

In this section, we design experiments to show the limitations of CFG and how CFG++ can effectively mitigate these downsides. The main experiments were conducted with SD v1.5 or SDXL with 50 NFE DDIM sampling. In this regime, we searched for the matching guidance values of $\omega$ and $\lambda$ for a fair comparison. We fix $\lambda = 0.2, 0.4, 0.6, 0.8, 1.0$ and find the $\omega$ values that produce the images that are of closest proximity in terms of LPIPS distance given the same seed. We found that the corresponding values were $\omega = 2.0, 5.0, 7.5, 9.0, 12.5$, respectively. Some of the experiments were also conducted with distilled model such as SDXL-turbo, lightning.

### 4.1 TEXT-TO-IMAGE GENERATION

Using the corresponding scales for $\omega$ and $\lambda$, we directly compare the performance of the T2I task using SD v1.5 and SDXL. In Tab. 1, we report quantitative metrics using 10k images generated from COCO captions (Lin et al., 2014). Here, we observe a constant improvement of the FID metric across all guidance scales (also see Fig. 15 for an apples-to-apples comparison), with approximately the same level of CLIP similarity or better. Additionally, we evaluate Im-

| Model | Metric | CFG | CFG++ (ours) |
|---|---|---|---|
| SDXL-Turbo | FID↓ | 59.67 | **59.21** |
| | CLIP↑ | 0.320 | **0.325** |
| | ImageReward↑ | 0.777 | **0.968** |
| SDXL-Lightning | FID↓ | 56.11 | **55.19** |
| | CLIP↑ | 0.322 | **0.324** |
| | ImageReward↑ | 0.691 | **0.829** |
| SD v1.5 (DPM++ 2M) | FID↓ | 32.72 | **32.58** |
| | CLIP↑ | **0.313** | 0.312 |
| | ImageReward↑ | **0.086** | 0.023 |

Table 2: Quant. eval. on accelerated T2I sampling

ageReward (IR) (Xu et al., 2024), trained on human judgment data for better correlation, and observe improvements in this metric as well. The improvements can also be clearly seen in Fig. 11 (SD v1.5), where the unnatural components of the generated images are corrected. Specifically, we see that unnatural depictions of human hands, and incorrect renderings of the text are corrected in CFG++, a

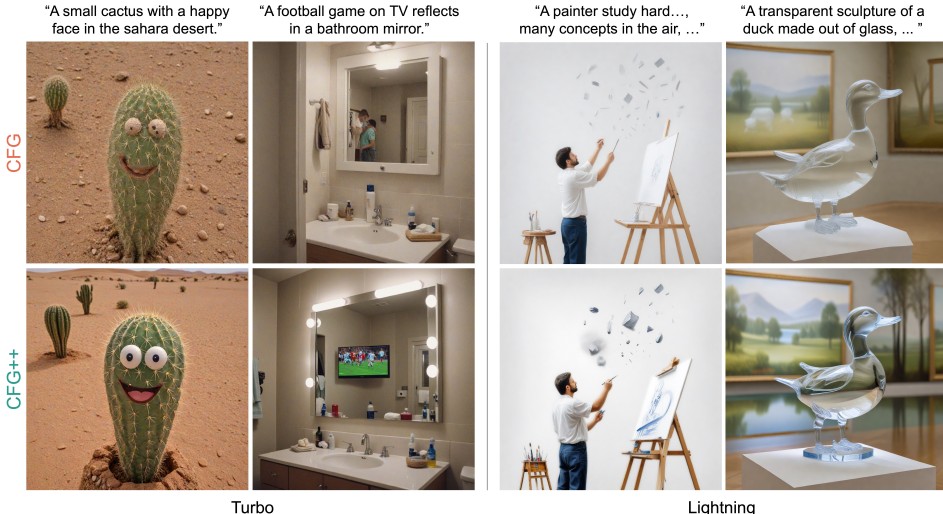

Figure 5: T2I using SDXL-{turbo, lightning}, 6 NFE, CFG vs CFG++.

long-standing research question in and of its own (Podell et al., 2023; Pelykh et al., 2024; Chen et al., 2023). We find that the improvement gain from CFG++ is even more dramatic for distilled diffusion models such as SDXL-{turbo, lightning}. We quantify the results on the same 5k prompts with 6 NFE sampling. In Fig. 5, Fig. 12 and Fig. 13 we see significant boosts in the quality of the generated images, which is also depicted in the improvements seen in Tab. 2.

To show the compatibility of CFG++ with higher-order solvers, we experiment with the DPM++ 2M (Lu et al., 2022b) solver using 20 NFE. We note that in such a low NFE regime, a CFG scale of 5.0 corresponds to a CFG++ scale of 1.0, and we have to slightly extrapolate the value of $\lambda \geq 1.0$ to achieve stronger guidance results. In Tab. 2, we observe that CFG++ is broadly comparable to CFG. This outcome differs from the consistent improvement of CFG++ over CFG observed with 50 NFE DDIM sampling. We attribute this to two factors: (1) the overall image quality for 20 NFE DPM++ 2M is generally lower compared to 50 NFE DDIM, leading to noisier quantitative metrics, particularly for metrics like ImageReward that are sensitive to subtle quality changes; and (2) the difference between CFG and CFG++ tends to be less pronounced in low NFE regimes without distillation, as stronger guidance effects emerge with higher NFEs.

## 4.2 DIFFUSION IMAGE INVERSION AND EDITING

We further explore the effect of CFG++ on DDIM inversion (Dhariwal & Nichol, 2021), where the source image is reverted to a latent vector that can reproduce the original image through generative process. DDIM inversion is well-known to break down in the usual CFG setting as CFG magnifies the accumulated error of each inversion step (Mokady et al., 2023), violating local linearization assumptions (Wallace et al., 2023). We show that CFG++ mitigates this issue by improving the inversion and image editing capabilities. We evaluate our method following the experimental setups in Park et al. (2024) and Kim et al. (2024b).

**Diffusion Image Inversion.** Using the matched set of scales for $\omega$ and $\lambda$, we demonstrate the effect of CFG++ on the diffusion image inversion task. Specifically, we reconstruct the images after inversion and evaluate it through PSNR and RMSE, following the methodology of Wallace et al. (2023). In Fig. 6, we illustrate the reconstructed examples (6a) from a real image from COCO data set and computed metrics (6b) for 5k COCO-2014 (Lin et al., 2014) validation set. Both qualitative and quantitative evaluations demonstrate a consistent improvement on reconstruction performance induced by CFG++. Notably, DDIM inversion with CFG++ leads to a consistent reconstruction of the source image across all guidance scales, while DDIM inversion with CFG fails to reconstruct it, as shown in Fig. 16 for another real image.

**Image Editing.** Fig. 6c compares the image editing result using CFG and CFG++ followed by image inversion with 5 NFEs. In the image editing stage, a word in the source text highlighted by green

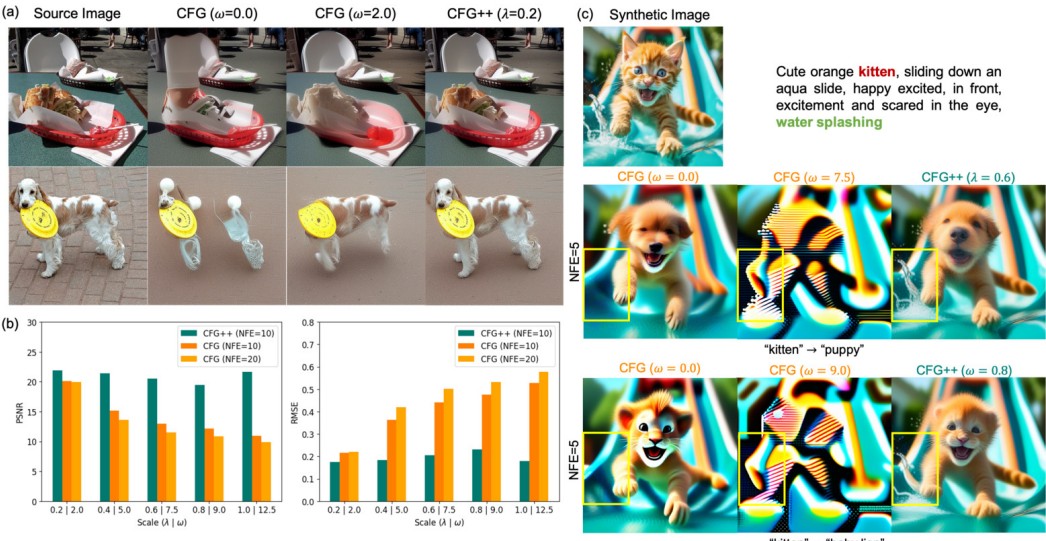

Figure 6: Inversion and editing results. (a) Reconstructed samples after inversion by CFG and CFG++. (b) Quantitative comparison between CFG and CFG++ for reconstruction. (c) Image editing comparison via SDXL.

color is swapped with the target concept, and this modified text is used as the condition for sampling. CFG++ successfully edits the target concept while preserving other concepts, such as background. In particular, the water splashing in the background, which tends to disappear in the conventional CFG, is maintained through the inversion process by CFG++. This emphasizes CFG++'s superior ability to retain specific scene elements that are frequently lost in the standard CFG approach. Moreover, standard CFG's disrupted DDIM inversion leads to saturated and less faithful editing results, whereas CFG++ delivers precise and high-fidelity edits.

## 4.3 TEXT-CONDITIONED INVERSE PROBLEMS

Inverse problem involves restoring the original data $\boldsymbol{x}$ from a noisy measurement $\boldsymbol{y} = \mathcal{A}(\boldsymbol{x}) + \boldsymbol{n}$, where $\mathcal{A}$ represents an imaging operator introduces distortions (e.g, Gaussian blur) and $\boldsymbol{n}$ denotes the measurement noise. Diffusion inverse solvers (DIS) address this challenge by leveraging pre-trained diffusion models as implicit priors and performing posterior sampling $\boldsymbol{x} \sim p(\boldsymbol{x}|\boldsymbol{y})$. Methods that leverage latent diffusion have gained recent interest (Rout et al., 2024; Song et al., 2024), but leveraging texts for solving these problems remains relatively underexplored (Chung et al., 2023b; Kim et al., 2023). One of the main reasons for this is that naively using CFG on top of latent DIS leads to diverging samples (Chung et al., 2023b). Several heuristic modifications such as null prompt optimization (Kim et al., 2023) with modified sampling schemes were needed to mitigate this drawback. This naturally leads to the question: Is it possible to leverage CFG guidance as a plug-and-play component of existing solvers? Here, we answer this question with a positive by showing that CFG++ enables the incorporation of text prompts into a standard solver. Specifically, we focus on comparing the performance of PSLD (Rout et al., 2024) combined with CFG and CFG++ in solving linear inverse problems. This evaluation utilizes the FFHQ (Karras et al., 2019) 512x512 dataset and the text prompt "a high-quality photo of a face". Further details about the experimental settings can be found in the Appendix E.

As shown in Tab. 3, our method mostly outperforms both the vanilla PSLD and PSLD with CFG. The superiority is also evident from the Fig. 7, where CFG++ consistently delivers high-quality reconstructions across all tasks. PSLD with CFG often suffer from artifacts and blurriness. Conversely, CFG++ achieves better fidelity, clearly distinguishing between faces and faithfully reproducing fine details like eyelids and hair texture. For more results, please refer to the Appendix F.

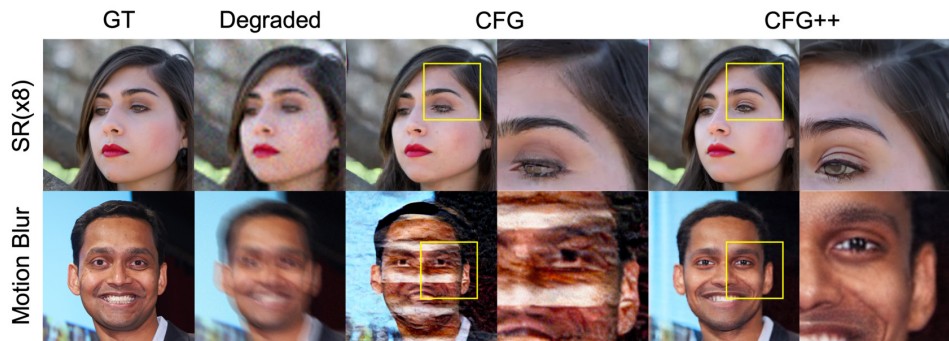

Figure 7: Qualitative comparison on various inverse problems using PSLD (Rout et al., 2024) under CFG and CFG++. For more results, please refer to the Appendix F.

| Method | SR (x8) | | | Deblur (motion) | | | Deblur (gauss) | | | Inpaint | | |
|---|---|---|---|---|---|---|---|---|---|---|---|---|
| | FID ↓ | LPIPS ↓ | PSNR ↑ | FID ↓ | LPIPS ↓ | PSNR ↑ | FID ↓ | LPIPS ↓ | PSNR ↑ | FID ↓ | LPIPS ↓ | PSNR ↑ |
| PSLD | 46.24 | 0.413 | 24.41 | 97.51 | 0.500 | 21.83 | 41.65 | **0.388** | 26.88 | 10.27 | 0.053 | 30.15 |
| PSLD + CFG | 41.24 | 0.394 | **24.91** | 91.90 | 0.493 | **22.29** | 41.52 | 0.390 | **26.94** | **9.36** | 0.055 | 30.27 |
| PSLD + CFG++ (ours) | **36.58** | **0.385** | 24.87 | **65.67** | **0.482** | 21.93 | **39.85** | 0.400 | 26.90 | 9.78 | **0.052** | **30.31** |

Table 3: Quantitative comparison (FID, LPIPS, PSNR) of PSLD, PSLD with CFG, and PSLD with CFG++ on Latent Diffusion Inverse Solver.

## 5 RELATED WORKS AND DISCUSSIONS

Kynkäänniemi et al. (2024) observed that applying CFG guidance at the earlier stages of sampling always led to detrimental effects and drastically reduced the diversity of the samples. The guidance at the later stages of sampling had minimal effects. Drawing upon these observations, it was empirically shown that applying CFG only in the limited time interval near the middle led to the best performance. Similar observations were made in the SD community (Howard & Prashanth, 2022; Birch, 2023) in adjusting the guidance scale across $t$. These works try to empirically adjust the strength of the guidance, while we derive our trajectory in a principled manner. More recently, Bradley & Nakkiran (2024) observed that the CFG score is not a valid denoising direction, showing that in the asymptotic limit, CFG sampling can be considered as a specific type of predictor-corrector (PC) sampling, dubbed PCG. PCG shares a similar spirit with CFG++ in that for the mainstream of sampling, one defers from the use of mixed CFG score (we use unconditional score, PCG uses conditional score) to avoid invalid directions. While PCG derives an interesting connection of PC sampling with CFG, the proposed algorithm does not improve the sampling performance. In contrast, CFG++ improves the sample quality without additional computation overhead. Since CFG++ attempts to make adjustments of CFG using a linear combination of score functions, it is not surprising to see that CFG++ sampling can be achieved by setting a time varying schedule. Specifically, by setting a time-varying schedule $\omega_t$ as $\omega_t = -\omega \left( \sqrt{1 - \bar{\alpha}_{t-1}} - \sqrt{\frac{1-\bar{\alpha}_{t-1}}{\alpha_t}} \right)$, one can achieve the same effect with CFG as CFG++. Details are shown in App. D. However, this specific choice of time-varying guidance scale has not been reported in the literature, as it is a schedule that is relatively complex to be drawn heuristically. We further show in Appendix C that such choice removes undesirable oscillatory behavior in the evolution of the posterior mean.

## 6 CONCLUSION

This paper revises the most widespread guidance method, classifier-free guidance (CFG), highlighting that a small and reasonable guidance scale, e.g. $\lambda \in [0.0, 1.0]$, might suffice successful guidance with a proper geometric correction. Observing that the original CFG suffers from off-manifold issues during sampling, we propose a simple but surprisingly effective fix. This change translates the conditional guidance from extrapolating to interpolating between unconditional and conditional sampling trajectories, leading to more interpretable guidance in contrast to the conventional CFG which relies on heuristic $\omega$ scaling. Given that CFG++ mitigates off-manifold issues, it may be beneficial for other downstream applications requiring accurate latent denoised estimate representations, e.g. DIS, as demonstrated in our experiments.

ACKNOWLEDGMENTS

This work was supported by the National Research Foundation of Korea under Grant RS-2024-00336454 and by the Institute for Information & Communications Technology Planning & Evaluation (IITP) grant funded by the Korea government (MSIT) (RS-2019-II190075, Artificial Intelligence Graduate School Program, KAIST).

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

| **Algorithm 3** DDIM Inversion with CFG | **Algorithm 4** DDIM Inversion with CFG++ |
|---|---|
| **Require:** $\boldsymbol{x}_0, 0 \le \omega \in \mathbb{R}$ | **Require:** $\boldsymbol{x}_0, \lambda \in [0,1]$ |
| 1: **for** $i = 0$ **to** $T-1$ **do** | 1: **for** $i = 0$ **to** $T-1$ **do** |
| 2: $\quad \hat{\boldsymbol{\epsilon}}_c^\omega(\boldsymbol{x}_t) = \hat{\boldsymbol{\epsilon}}_\varnothing(\boldsymbol{x}_t) + \omega[\hat{\boldsymbol{\epsilon}}_c(\boldsymbol{x}_t) - \hat{\boldsymbol{\epsilon}}_\varnothing(\boldsymbol{x}_t)]$ | 2: $\quad \hat{\boldsymbol{\epsilon}}_c^\lambda(\boldsymbol{x}_t) = \hat{\boldsymbol{\epsilon}}_\varnothing(\boldsymbol{x}_t) + \lambda[\hat{\boldsymbol{\epsilon}}_c(\boldsymbol{x}_t) - \hat{\boldsymbol{\epsilon}}_\varnothing(\boldsymbol{x}_t)]$ |
| 3: $\quad \hat{\boldsymbol{x}}_c^\omega(\boldsymbol{x}_t) \leftarrow (\boldsymbol{x}_t - \sqrt{1-\bar{\alpha}_t}\hat{\boldsymbol{\epsilon}}_c^\omega(\boldsymbol{x}_t))/\sqrt{\bar{\alpha}_t}$ | 3: $\quad \hat{\boldsymbol{x}}_c^\lambda(\boldsymbol{x}_t) \leftarrow (\boldsymbol{x}_t - \sqrt{1-\bar{\alpha}_t}\hat{\boldsymbol{\epsilon}}_\varnothing(\boldsymbol{x}_t))/\sqrt{\bar{\alpha}_t}$ |
| 4: $\quad \boldsymbol{x}_{t+1} = \sqrt{\bar{\alpha}_{t+1}}\hat{\boldsymbol{x}}_c^\omega(\boldsymbol{x}_t) + \sqrt{1-\bar{\alpha}_{t+1}}\hat{\boldsymbol{\epsilon}}_c^\omega(\boldsymbol{x}_t)$ | 4: $\quad \boldsymbol{x}_{t+1} = \sqrt{\bar{\alpha}_{t+1}}\hat{\boldsymbol{x}}_c^\lambda(\boldsymbol{x}_t) + \sqrt{1-\bar{\alpha}_{t+1}}\hat{\boldsymbol{\epsilon}}_c^\lambda(\boldsymbol{x}_t)$ |
| 5: **end for** | 5: **end for** |
| 6: **return** $\boldsymbol{x}_T$ | 6: **return** $\boldsymbol{x}_T$ |

Figure 8: Comparison between DDIM inversion. CFG++ proposes a simple yet effective fix: using $\hat{\boldsymbol{\epsilon}}_\varnothing(\boldsymbol{x}_t)$ instead of $\hat{\boldsymbol{\epsilon}}_c^\omega(\boldsymbol{x}_t)$ in Tweedie's denoising step.

## A    EXTENSION OF CFG++ TO OTHER SOLVERS

In this section, we discuss ways in which we can apply CFG++ to a more diverse set of ODE/SDE solvers. We consider solving the variance exploding (VE) PF-ODE as presented in (1), as reparametrization VP diffusion models can easily recover the VE formulation, as often implemented in widely used frameworks such as `https://github.com/crowsonkb/k-diffusion`. Following the notation in Lu et al. (2022b), we consider a sequence of timesteps $\{t_i\}_{i=0}^M$, where $t_0 = T$ denotes the initial starting point of the reverse sampling (i.e. Gaussian noise).

**Euler (Karras et al., 2022)**    The construction is the same as in DDIM. We include it here for completeness. The update step reads

$$\boldsymbol{x}_{t_{i+1}} = \hat{\boldsymbol{x}}_c^\omega(\boldsymbol{x}_{t_i}) + \frac{\boldsymbol{x}_{t_i} - \hat{\boldsymbol{x}}_c^\omega(\boldsymbol{x}_{t_i})}{\sigma_{t_i}} \cdot \sigma_{t_{i+1}}, \tag{CFG}$$

$$\boldsymbol{x}_{t_{i+1}} = \hat{\boldsymbol{x}}_c^\lambda(\boldsymbol{x}_{t_i}) + \frac{\boldsymbol{x}_{t_i} - \hat{\boldsymbol{x}}_\varnothing(\boldsymbol{x}_{t_i})}{\sigma_{t_i}} \cdot \sigma_{t_{i+1}}, \tag{CFG++}$$

**Euler Ancestral**    Euler Ancestral sampler follows Euler sampler, but introduces stochasticity by taking larger steps and then adding a slight amount of noise. Adjusting with CFG++ is straightforward.

$$\boldsymbol{x}_{t_{i+1}} = \hat{\boldsymbol{x}}_c^\omega(\boldsymbol{x}_{t_i}) + \frac{\boldsymbol{x}_{t_i} - \hat{\boldsymbol{x}}_c^\omega(\boldsymbol{x}_{t_i})}{\sigma_{t_i}} \cdot (\sigma_{t_{d_i}} - \sigma_{t_i}) + \sigma_{t_i}\boldsymbol{\epsilon}, \tag{CFG}$$

$$\boldsymbol{x}_{t_{i+1}} = \hat{\boldsymbol{x}}_c^\lambda(\boldsymbol{x}_{t_i}) + \frac{\boldsymbol{x}_{t_i} - \hat{\boldsymbol{x}}_\varnothing(\boldsymbol{x}_{t_i})}{\sigma_{t_i}} \cdot (\sigma_{t_{d_i}} - \sigma_{t_i}) + \sigma_{t_i}\boldsymbol{\epsilon}, \tag{CFG++}$$

where $t_i > t_{d_i} > t_{i+1}$ and $\boldsymbol{\epsilon} \sim \mathcal{N}(0, \boldsymbol{I})$.

**DPM-solver++ 2M (Lu et al., 2022b)**    Define $\sigma_t := e^{-t}$, $h_i := t_i - t_{i-1}$, and $r_i := h_{i-1}/h_i$. After initializing $\boldsymbol{x}_{t_0}$ with Gaussian noise, the first iteration is given by

$$\boldsymbol{x}_{t_1} = \hat{\boldsymbol{x}}_c^\omega(\boldsymbol{x}_{t_0}) + e^{-h_1}(\boldsymbol{x}_{t_0} - \hat{\boldsymbol{x}}_c^\omega(\boldsymbol{x}_{t_0})) \tag{CFG}$$

$$\boldsymbol{x}_{t_1} = \hat{\boldsymbol{x}}_c^\lambda(\boldsymbol{x}_{t_0}) + e^{-h_1}(\boldsymbol{x}_{t_0} - \hat{\boldsymbol{x}}_\varnothing(\boldsymbol{x}_{t_0})) \tag{CFG++}$$

The following iterations by using the standard CFG reads

$$\boldsymbol{D}_i = \hat{\boldsymbol{x}}_c^\omega(\boldsymbol{x}_{t_{i-1}}) + \frac{1}{2r_i}\left(\hat{\boldsymbol{x}}_c^\omega(\boldsymbol{x}_{t_{i-1}}) - \hat{\boldsymbol{x}}_c^\omega(\boldsymbol{x}_{t_{i-2}})\right), \tag{23}$$

$$\boldsymbol{x}_{t_i} = e^{-h_i}\boldsymbol{x}_{t_{i-1}} - (e^{-h_i}-1)\boldsymbol{D}_i. \tag{24}$$

Rearranging (23), (24), we can rewrite the update steps as

$$\boldsymbol{x}_{t_i} = \hat{\boldsymbol{x}}_c^\omega(\boldsymbol{x}_{t_{i-1}}) - e^{-h_i}\hat{\boldsymbol{x}}_c^\omega(\boldsymbol{x}_{t_{i-1}}) + \frac{1-e^{-h_i}}{2r_i}\left(\hat{\boldsymbol{x}}_c^\omega(\boldsymbol{x}_{t_{i-1}}) - \hat{\boldsymbol{x}}_c^\omega(\boldsymbol{x}_{t_{i-2}})\right) + e^{-h_i}\boldsymbol{x}_{t_{i-1}}. \tag{25}$$

Notice that in order to apply CFG++ to (25), we should only keep the first term as the conditional Tweedie, and use the unconditional estimates for the rest of the components. i.e.

$$\boldsymbol{x}_{t_i} = \hat{\boldsymbol{x}}_c^\lambda(\boldsymbol{x}_{t_{i-1}}) - e^{-h_i}\hat{\boldsymbol{x}}_\varnothing(\boldsymbol{x}_{t_{i-1}}) + \frac{1-e^{-h_i}}{2r_i}\left(\hat{\boldsymbol{x}}_\varnothing(\boldsymbol{x}_{t_{i-1}}) - \hat{\boldsymbol{x}}_\varnothing(\boldsymbol{x}_{t_{i-2}})\right) + e^{-h_i}\boldsymbol{x}_{t_{i-1}} \tag{26}$$

**DPM-solver++ 2S (Lu et al., 2022b)** With the same choices of $\sigma_t, h_i$, we additionally define the timesteps $\{s_i\}_{i=1}^M$ with $t_i > s_{i+1} > t_{i+1}$. Further, let $r_i = \frac{s_i - t_{i-1}}{t_i - t_{i-1}}$. Using standard CFG, the iteration reads

$$\boldsymbol{u}_i = e^{-r_i h_i}\boldsymbol{x}_{t_{i-1}} + (1 - e^{-r_i h_i})\hat{\boldsymbol{x}}_{\boldsymbol{c}}^\omega(\boldsymbol{x}_{t_{i-1}}) \tag{27}$$

$$\boldsymbol{x}_{t_i} = \hat{\boldsymbol{x}}_{\boldsymbol{c}}^\omega(\boldsymbol{x}_{t_{i-1}}) - e^{-h_i}\hat{\boldsymbol{x}}_{\boldsymbol{c}}^\omega(\boldsymbol{x}_{t_{i-1}}) + \frac{1 - e^{-h_i}}{2r_i}\left(\hat{\boldsymbol{x}}_{\boldsymbol{c}}^\omega(\boldsymbol{u}_i) - \hat{\boldsymbol{x}}_{\boldsymbol{c}}^\omega(\boldsymbol{x}_{t_{i-1}})\right) + e^{-h_i}\boldsymbol{x}_{t_{i-1}} \tag{28}$$

Applying the general transition rule from CFG to CFG++ as introduced in (15), we can keep all of the Tweedie estimates to be unconditional, and only change the first term of (28), i.e.

$$\boldsymbol{u}_i = e^{-r_i h_i}\boldsymbol{x}_{t_{i-1}} + (1 - e^{-r_i h_i})\hat{\boldsymbol{x}}_\varnothing(\boldsymbol{x}_{t_{i-1}}) \tag{29}$$

$$\boldsymbol{x}_{t_i} = \hat{\boldsymbol{x}}_\varnothing(\boldsymbol{x}_{t_{i-1}}) - e^{-h_i}\hat{\boldsymbol{x}}_\varnothing(\boldsymbol{x}_{t_{i-1}}) + \frac{1 - e^{-h_i}}{2r_i}\left(\hat{\boldsymbol{x}}_{\boldsymbol{c}}^\lambda(\boldsymbol{u}_i) - \hat{\boldsymbol{x}}_\varnothing(\boldsymbol{x}_{t_{i-1}})\right) + e^{-h_i}\boldsymbol{x}_{t_{i-1}}. \tag{30}$$

For the ancestral version of DPM-solver++ 2S, we can follow the general transition rule, as was shown for Euler $\to$ Euler Ancestral.

## B   Extension of CFG++ to flow matching

Here, we generalize CFG++ with flow matching which subsumes diffusion models as specific instantiations. Let $p_0$ and $p_1$ be two data distributions on $\mathbb{R}^n$, and $Q_{01}$ denotes a coupling of $p_0$ and $p_1$. Then, the flow matching loss is defined as follows:

$$\ell_{\text{FM}}(\theta; Q_{01}) = \mathbb{E}_{(\boldsymbol{x}_0, \boldsymbol{x}_1) \sim Q_{01}, t \sim \text{unif}(0,1)}[\ell_{\text{MSE}}(\boldsymbol{x}_1 - \boldsymbol{x}_0, \boldsymbol{v}_\theta(\boldsymbol{x}_t, t))], \tag{31}$$

where $\boldsymbol{v}_\theta : \mathbb{R}^n \times (0,1) \to \mathbb{R}^n$ refers to a velocity parameterized by $\theta$, $\ell_{\text{MSE}}(\boldsymbol{x}, \boldsymbol{y}) = \|\boldsymbol{x} - \boldsymbol{y}\|_2^2$, and $\boldsymbol{x}_t = (1 - t)\boldsymbol{x}_0 + t\boldsymbol{x}_1$. This objective can be reformulated in the form of denoising as follows:

$$\ell_{\text{FM}}(\theta; Q_{01}) = \mathbb{E}_{(\boldsymbol{x}_0, \boldsymbol{x}_1) \sim Q_{01}, t \sim \text{unif}(0,1)}\left[\frac{1}{t^2}\ell_{\text{MSE}}(\boldsymbol{x}_0, \boldsymbol{D}_\theta(\boldsymbol{x}_t, t))\right], \tag{32}$$

where $\boldsymbol{D}_\theta(\boldsymbol{x}_t, t) = \boldsymbol{x}_t - t\boldsymbol{v}_\theta(\boldsymbol{x}_t, t)$ serves as a denoiser function. Then, one can derive an ODE which translates samples from $p_0$ and $p_1$ given a vector field $\boldsymbol{v}_\theta$:

$$d\boldsymbol{x}_t = \boldsymbol{v}_\theta(\boldsymbol{x}_t, t)dt = \frac{\boldsymbol{x}_t - \boldsymbol{D}_\theta(\boldsymbol{x}_t, t)}{t}dt, \quad t \in (0,1), \tag{33}$$

which resembles (1) with a choice of $\sigma_t = t$. For text-conditional flow, CFG can be applied to (33) by solving the following ODE (Kim et al., 2024a):

$$d\boldsymbol{x}_t = \left[\boldsymbol{v}_\theta(\boldsymbol{x}_t, t, \varnothing) + \omega(\boldsymbol{v}_\theta(\boldsymbol{x}_t, t, \boldsymbol{c}) - \boldsymbol{v}_\theta(\boldsymbol{x}_t, t, \varnothing))\right]dt = \tilde{\boldsymbol{v}}_{\boldsymbol{c}}^\omega(\boldsymbol{x}_t)dt, \quad t \in (0,1). \tag{34}$$

We similarly define $\tilde{\boldsymbol{x}}_{\boldsymbol{c}}^\omega(\boldsymbol{x}_t) = \boldsymbol{x}_t - t\tilde{\boldsymbol{v}}_{\boldsymbol{c}}^\omega(\boldsymbol{x}_t)$, $\tilde{\boldsymbol{x}}_\varnothing(\boldsymbol{x}_t) = \boldsymbol{x}_t - t\boldsymbol{v}_\theta(\boldsymbol{x}_t, t, \varnothing) = \boldsymbol{x}_t - t\tilde{\boldsymbol{v}}_\varnothing(\boldsymbol{x}_t)$, and $\tilde{\boldsymbol{x}}_{\boldsymbol{c}}^\lambda(\boldsymbol{x}_t) = \boldsymbol{x}_t - t\tilde{\boldsymbol{v}}_{\boldsymbol{c}}^\lambda(\boldsymbol{x}_t)$. Based on this ODE, we can reproduce results from a diverse set of solvers in Sec. A. For example, a single update step of Euler solver reads:

$$\boldsymbol{x}_{t_{i+1}} = \tilde{\boldsymbol{x}}_{\boldsymbol{c}}^\omega(\boldsymbol{x}_{t_i}) + \frac{\boldsymbol{x}_{t_i} - \tilde{\boldsymbol{x}}_{\boldsymbol{c}}^\omega(\boldsymbol{x}_{t_i})}{t_i} \cdot t_{i+1}, \tag{CFG}$$

$$\boldsymbol{x}_{t_{i+1}} = \tilde{\boldsymbol{x}}_{\boldsymbol{c}}^\lambda(\boldsymbol{x}_{t_i}) + \frac{\boldsymbol{x}_{t_i} - \tilde{\boldsymbol{x}}_\varnothing(\boldsymbol{x}_{t_i})}{t_i} \cdot t_{i+1}. \tag{CFG++}$$

This implies that CFG++ is potentially compatible with various flow-based generative models.

## C   Evolution of the posterior mean: CFG vs. CFG++

Recall that one can equivalently view the evolution of the posterior mean $\mathbb{E}[\boldsymbol{x}_0|\boldsymbol{x}_t]$ rather than the noisy variables $\boldsymbol{x}_t$. In this context, we derive the sequential evolution of conditional posterior mean $\hat{\boldsymbol{x}}_{\boldsymbol{c}}^\omega$ and $\hat{\boldsymbol{x}}_{\boldsymbol{c}}^\lambda$ through time $t$ to further understand the underlying behavior of the proposed sampling.

**Proposition 1.** *Let $d\boldsymbol{z}(\boldsymbol{x}_t) := \boldsymbol{z}(\boldsymbol{x}_t) - \boldsymbol{z}(\boldsymbol{x}_{t+1})$ denote the discrete time evolution of some random variable $\boldsymbol{z}$ at time $t$. Then, the evolution of $\hat{\boldsymbol{x}}_{\boldsymbol{c}}^{\omega}$ of CFG and $\hat{\boldsymbol{x}}_{\boldsymbol{c}}^{\lambda}$ of CFG++ is given by*

$$d\hat{\boldsymbol{x}}_{\boldsymbol{c}}^{\omega}(\boldsymbol{x}_t) = \frac{\sqrt{1 - \bar{\alpha}_t}}{\sqrt{\bar{\alpha}_t}} d\hat{\boldsymbol{\epsilon}}_{\varnothing}(\boldsymbol{x}_t) + \omega(\Delta(\boldsymbol{x}_t, \boldsymbol{c}) - \Delta(\boldsymbol{x}_{t+1}, \boldsymbol{c})) \tag{35}$$

$$d\hat{\boldsymbol{x}}_{\boldsymbol{c}}^{\lambda}(\boldsymbol{x}_t) = \frac{\sqrt{1 - \bar{\alpha}_t}}{\sqrt{\bar{\alpha}_t}} \underbrace{d\hat{\boldsymbol{\epsilon}}_{\varnothing}(\boldsymbol{x}_t)}_{\text{uncond. shift}} + \lambda \underbrace{\Delta(\boldsymbol{x}_t, \boldsymbol{c})}_{\text{cond. shift}}, \tag{36}$$

*where $\Delta(\boldsymbol{x}_t, \boldsymbol{c}) := \hat{\boldsymbol{x}}_{\boldsymbol{c}}(\boldsymbol{x}_t) - \hat{\boldsymbol{x}}_{\varnothing}(\boldsymbol{x}_t)$.*

*Proof.* We start by writing the iteration from $t + 1 \rightarrow t$

$$\boldsymbol{x}_t = \sqrt{\bar{\alpha}_t}\hat{\boldsymbol{x}}_{\boldsymbol{c}}^{\omega}(\boldsymbol{x}_{t+1}) + \sqrt{1 + \bar{\alpha}_t}\hat{\boldsymbol{\epsilon}}_{\boldsymbol{c}}^{\omega}(\boldsymbol{x}_{t+1}). \tag{37}$$

The Tweedie estimate for the next step is then written as

$$\hat{\boldsymbol{x}}_{\boldsymbol{c}}^{\omega}(\boldsymbol{x}_t) = \frac{\boldsymbol{x}_t - \sqrt{1 - \bar{\alpha}_t}\hat{\boldsymbol{\epsilon}}_{\boldsymbol{c}}^{\omega}(\boldsymbol{x}_t)}{\sqrt{\bar{\alpha}_t}} \tag{38}$$

$$= \frac{\sqrt{\bar{\alpha}_t}\hat{\boldsymbol{x}}_{\boldsymbol{c}}^{\omega}(\boldsymbol{x}_{t+1}) + \sqrt{1 - \bar{\alpha}_t}(\hat{\boldsymbol{\epsilon}}_{\boldsymbol{c}}^{\omega}(\boldsymbol{x}_{t+1}) - \hat{\boldsymbol{\epsilon}}_{\boldsymbol{c}}^{\omega}(\boldsymbol{x}_t))}{\sqrt{\bar{\alpha}_t}} \tag{39}$$

$$= \hat{\boldsymbol{x}}_{\boldsymbol{c}}^{\omega}(\boldsymbol{x}_{t+1}) + \frac{\sqrt{1 - \bar{\alpha}_t}}{\sqrt{\bar{\alpha}_t}} \Big[ \hat{\boldsymbol{\epsilon}}_{\varnothing}(\boldsymbol{x}_{t+1}) - \hat{\boldsymbol{\epsilon}}_{\varnothing}(\boldsymbol{x}_t) + \omega(\hat{\boldsymbol{\epsilon}}_{\boldsymbol{c}}(\boldsymbol{x}_{t+1}) - \hat{\boldsymbol{\epsilon}}_{\varnothing}(\boldsymbol{x}_{t+1}))$$
$$- \omega(\hat{\boldsymbol{\epsilon}}_{\boldsymbol{c}}(\boldsymbol{x}_t) - \hat{\boldsymbol{\epsilon}}_{\varnothing}(\boldsymbol{x}_t)) \Big]. \tag{40}$$

Using the relation $\hat{\boldsymbol{\epsilon}}_{\boldsymbol{c}}^{\omega}(\boldsymbol{x}_t) = -(\sqrt{\bar{\alpha}_t}\hat{\boldsymbol{x}}_{\boldsymbol{c}}^{\omega}(\boldsymbol{x}_t) - \boldsymbol{x}_t)/\sqrt{1 - \bar{\alpha}_t}$, we have

$$d\hat{\boldsymbol{x}}_{\boldsymbol{c}}^{\omega}(\boldsymbol{x}_t) = \frac{\sqrt{1 - \bar{\alpha}_t}}{\sqrt{\bar{\alpha}_t}} d\hat{\boldsymbol{\epsilon}}_{\varnothing}(\boldsymbol{x}_t) + \omega \left( \hat{\boldsymbol{x}}_{\boldsymbol{c}}(\boldsymbol{x}_t) - \hat{\boldsymbol{x}}_{\varnothing}(\boldsymbol{x}_t) \right) - \omega \left( \hat{\boldsymbol{x}}_{\boldsymbol{c}}(\boldsymbol{x}_{t+1}) - \hat{\boldsymbol{x}}_{\varnothing}(\boldsymbol{x}_{t+1}) \right). \tag{41}$$

Similarly, for CFG++,

$$\hat{\boldsymbol{x}}_{\boldsymbol{c}}^{\lambda}(\boldsymbol{x}_t) = \frac{\boldsymbol{x}_t - \sqrt{1 - \bar{\alpha}_t}\hat{\boldsymbol{\epsilon}}_{\boldsymbol{c}}^{\lambda}(\boldsymbol{x}_t)}{\sqrt{\bar{\alpha}_t}} \tag{42}$$

$$= \frac{\sqrt{\bar{\alpha}_t}\hat{\boldsymbol{x}}_{\boldsymbol{c}}^{\lambda}(\boldsymbol{x}_{t+1}) + \sqrt{1 - \bar{\alpha}_t}(\hat{\boldsymbol{\epsilon}}_{\varnothing}(\boldsymbol{x}_{t+1}) - \hat{\boldsymbol{\epsilon}}_{\boldsymbol{c}}^{\lambda}(\boldsymbol{x}_t))}{\sqrt{\bar{\alpha}_t}} \tag{43}$$

$$= \hat{\boldsymbol{x}}_{\boldsymbol{c}}^{\lambda}(\boldsymbol{x}_{t+1}) + \frac{\sqrt{1 - \bar{\alpha}_t}}{\sqrt{\bar{\alpha}_t}} \left[ \hat{\boldsymbol{\epsilon}}_{\varnothing}(\boldsymbol{x}_{t+1}) - \hat{\boldsymbol{\epsilon}}_{\varnothing}(\boldsymbol{x}_t) - \lambda(\hat{\boldsymbol{\epsilon}}_{\boldsymbol{c}}(\boldsymbol{x}_t) - \hat{\boldsymbol{\epsilon}}_{\varnothing}(\boldsymbol{x}_t)) \right]. \tag{44}$$

Hence,

$$d\hat{\boldsymbol{x}}_{\boldsymbol{c}}^{\lambda}(\boldsymbol{x}_t) = \frac{\sqrt{1 - \bar{\alpha}_t}}{\sqrt{\bar{\alpha}_t}} d\hat{\boldsymbol{\epsilon}}_{\varnothing}(\boldsymbol{x}_t) + \lambda(\hat{\boldsymbol{x}}_{\boldsymbol{c}}(\boldsymbol{x}_t) - \hat{\boldsymbol{x}}_{\varnothing}(\boldsymbol{x}_t)), \tag{45}$$

$\square$

Proposition 1 implies that the CFG++ update of $\hat{\boldsymbol{x}}_{\boldsymbol{c}}^{\lambda}$ is decomposed into two shift terms: 1) $d\hat{\boldsymbol{\epsilon}}_{\varnothing}(\boldsymbol{x}_t)$ represents the difference between consecutive unconditional scores (i.e. unconditional shift), and 2) $\Delta(\boldsymbol{x}_t, \boldsymbol{c})$ denotes the direction of conditional guidance at time $t$ (i.e. conditional shift). The conditional shift term is multiplied by a small interpolation factor $\lambda$ in the case of CFG++, inducing a small nudge toward the condition.

CFG $\hat{\boldsymbol{x}}_{\boldsymbol{c}}^{\omega}$, on the other hand, has the same unconditional shift, but has an **oscillatory behavior** for the conditional shift. The difference between CFG/CFG++ sampling arises from the unexpected additional shift from the previous timestep $t + 1$ that exists for the CFG decomposition: $-\Delta(\boldsymbol{x}_{t+1}, \boldsymbol{c})$, and the scaling constant $\omega$. The initial conditional shift $\omega\Delta(\boldsymbol{x}_t, \boldsymbol{c})$ with a large $\omega$ pushes the trajectory off the manifold, but cancels some of its effects by subtracting the conditional shift from the previous step $\omega\Delta(\boldsymbol{x}_{t+1}, \boldsymbol{c})$. The compounded vector $\Delta(\boldsymbol{x}_t, \boldsymbol{c}) - \Delta(\boldsymbol{x}_{t+1}, \boldsymbol{c})$ does induce a nudge closer to the condition but requires a large value of $\omega$ to have a meaningful effect, and thus is hard to interpret.

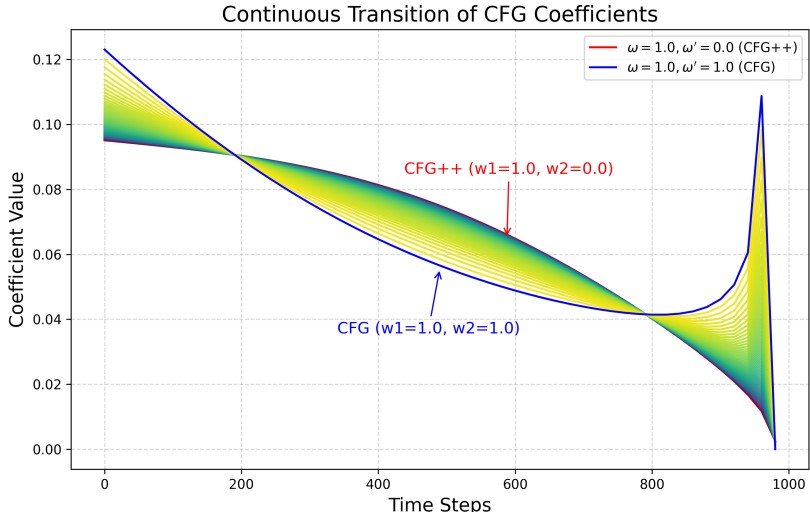

Figure 9: Continuous transition between CFG and CFG++ schedule

## D  CFG++ AS REWEIGHTED CFG

One can rewrite the update step of CFG and CFG++ succinctly as

$$\boldsymbol{x}_{t-1} = \sqrt{\bar{\alpha}_{t-1}}\hat{\boldsymbol{x}}_{\boldsymbol{c}}^{\omega} + \sqrt{1-\bar{\alpha}_{t-1}}\hat{\boldsymbol{\epsilon}}_{\boldsymbol{c}}^{\omega'} \tag{46}$$

$$= \sqrt{\bar{\alpha}_{t-1}}\left(\frac{\boldsymbol{x}_t - \sqrt{1-\bar{\alpha}_t}\hat{\boldsymbol{\epsilon}}_{\boldsymbol{c}}^{\omega}(\boldsymbol{x}_t)}{\sqrt{\bar{\alpha}_t}} + \sqrt{1-\bar{\alpha}_{t-1}}\hat{\boldsymbol{\epsilon}}_{\boldsymbol{c}}^{\omega'}\right), \tag{47}$$

where for CFG, $\omega' = \omega$, and for CFG++, $\omega' = 0$. Note that we are using the same notation $\omega$ here, but the guidance scale (represented as $\lambda$ in the previous sections), is much smaller. One can derive

$$\boldsymbol{x}_{t-1} = \sqrt{\bar{\alpha}_{t-1}}\hat{\boldsymbol{x}}_{\varnothing} - \frac{\omega\sqrt{1-\bar{\alpha}_t}(\hat{\boldsymbol{\epsilon}}_{\boldsymbol{c}} - \hat{\boldsymbol{\epsilon}}_{\varnothing})}{\sqrt{\alpha_t}} + \sqrt{1-\bar{\alpha}_{t-1}}\hat{\boldsymbol{\epsilon}}_{\varnothing} + \sqrt{1-\bar{\alpha}_{t-1}}\omega'(\hat{\boldsymbol{\epsilon}}_{\boldsymbol{c}} - \hat{\boldsymbol{\epsilon}}_{\varnothing}) \tag{48}$$

$$= \left(\sqrt{\bar{\alpha}_{t-1}}\hat{\boldsymbol{x}}_{\varnothing} + \sqrt{1-\bar{\alpha}_{t-1}}\hat{\boldsymbol{\epsilon}}_{\varnothing}\right) - \frac{\omega\sqrt{1-\bar{\alpha}_t}(\hat{\boldsymbol{\epsilon}}_{\boldsymbol{c}} - \hat{\boldsymbol{\epsilon}}_{\varnothing})}{\sqrt{\alpha_t}} + \sqrt{1-\bar{\alpha}_{t-1}}\omega'(\hat{\boldsymbol{\epsilon}}_{\boldsymbol{c}} - \hat{\boldsymbol{\epsilon}}_{\varnothing}) \tag{49}$$

$$= \boldsymbol{x}_{t-1,\varnothing} + (\hat{\boldsymbol{\epsilon}}_{\boldsymbol{c}} - \hat{\boldsymbol{\epsilon}}_{\varnothing})\underbrace{\left(\omega'\sqrt{1-\bar{\alpha}_{t-1}} - \omega\sqrt{\frac{1-\bar{\alpha}_{t-1}}{\alpha_t}}\right)}_{=:-\omega_t} \tag{50}$$

Here, we see that the guidance scale used in the original CFG is a composition of two time-dependent functions. When composed with the same strength for $\omega'$ and $\omega$, as depicted in Fig. 9 (b), the guidance scale of CFG peaks at the earlier stages of sampling, drops down, then rises back up. The sudden peak in the earlier stages may explain the unnatural saturation in the earlier stages of sampling using CFG. In contrast, the guidance scale of CFG++ has a convex-like function, where the scale gradually increases to some value.

By enforcing that the guidance scales integrate to 1 (i.e. enforce it as a PDF), we can generalize CFG/CFG++, and scrutinize the behavior when we interpolate between these two by setting $0 < \omega' < \omega$. The guidance scale functions for these interpolations are shown in Fig. 9. In Tab. 4, we see that the image quality measured by FID and ImageReward on COCO-1K data, gradually gets better as we transition from CFG++ to CFG.

| | $\omega' = 0.0$ (**CFG++**) | $\omega' = 0.4$ | $\omega' = 0.6$ | $\omega' = 0.8$ | $\omega' = 1.0$ (**CFG**) |
|---|---|---|---|---|---|
| ImageReward $\uparrow$ | **-0.112** | -0.125 | -0.126 | -0.153 | -0.420 |
| FID $\downarrow$ | **66.52** | 66.60 | 66.78 | 67.50 | 68.10 |

Table 4: Quantitative results of COCO-1k by interpolating between CFG and CFG++.

# E  EXPERIMENTAL DETAILS

## E.1  TEXT-CONDITIONED INVERSE PROBLEMS

**Problem settings.** We evaluate our approach across following degradation types: 1) Super-resolution with a scale factor of x8, 2) Motion deblurring from an image convolved with a 61 x 61 motion kernel, randomly sampled with an intensity value $0.3^2$, 3) Gaussian deblurring from an image convolved with a 61 x 61 kernel with an intensity value 0.5, and 4) Inpainting from 10-20% free-form masking, as implemented in (Saharia et al., 2022). In inpainting evaluations, regions outside the mask are overlaid with the ground truth image.

**Datasets, Models.** For evaluation, we use the FFHQ (Karras et al., 2019) 512x512 dataset and follow (Chung et al., 2023a) by selecting the first 1,000 images for testing. For the pre-trained latent diffusion model including baseline methods, we choose SD v1.5 trained on the LAION dataset. As a baseline for latent DIS, we consider PSLD (Rout et al., 2024). It enforces fidelity by projecting onto the subspace of $\mathcal{A}$ during the intermediate step between decoding and encoding, in conjunction with the DPS (Chung et al., 2023a) loss term. For gradient updates in both vanilla PSLD and PSLD with CFG, we use static step sizes of $\eta = 1.0$ and $\gamma = 0.1$ as recommended in (Rout et al., 2024). For CFG scale $\omega$, we applied the corresponding scales that we found to be corresponded for CFG++ scale $\lambda$. Please refer to the Tab. 5 for the hyperparameters used for PSLD with CFG++.

| | SR(x8) | Deblur(motion) | Deblur(gauss) | Inpaint |
|---|---|---|---|---|
| $\eta$ | 1.3 | 0.4 | 1.0 | 1.0 |
| $\gamma$ | 0.1 | 0.025 | 0.12 | 0.1 |
| $\lambda$ | 0.1 | 0.2 | 0.2 | 0.6 |
| $\omega$ | 1.5 | 2.0 | 2.0 | 7.5 |

Table 5: Hyperparameters for PSLD (Rout et al., 2024) with CFG++ and corresponding CFG scale $\omega$.

# F  FURTHER EXPERIMENTAL RESULTS

## F.1  T2I

Since the range of guidance scales for CFG and CFG++ is different, we matched the guidance values of $\omega$ and $\lambda$ for comparison by computing the LPIPS distance using the same seed.

Fig. 14 shows the clean estimates at each diffusion sampling time $t$, computed using Tweedie's formula, both with and without CFG. It is evident that the original CFG induces significant error, particularly at earlier times, leading to off-manifold samples. However, the CFG++ addresses this issue by adjusting the DDIM re-noising step and the guidance scale.

## F.2  REAL IMAGE EDITING

In Fig. 17-19, we provide qualitative comparison on real image editing via DDIM inversion with CFG and CFG++. For the DDIM inversion stage, we use "a photography of [source concept]" as conditioning prompt. For the sampling stage, we swap [source concept] to [target concept] and use it as conditioning prompt for generation. For example, in Fig. 17, we set [source concept] to "dog" and [target concept] to "cat". For all experiments, we set the guidance scale as $\omega = 9.0$ and $\lambda = 0.8$ as described in the main paper. The comparison demonstrates that CFG++ successfully edits the given image which was not possible by CFG. This results also support our claim on reduced error during DDIM inversion by CFG++.

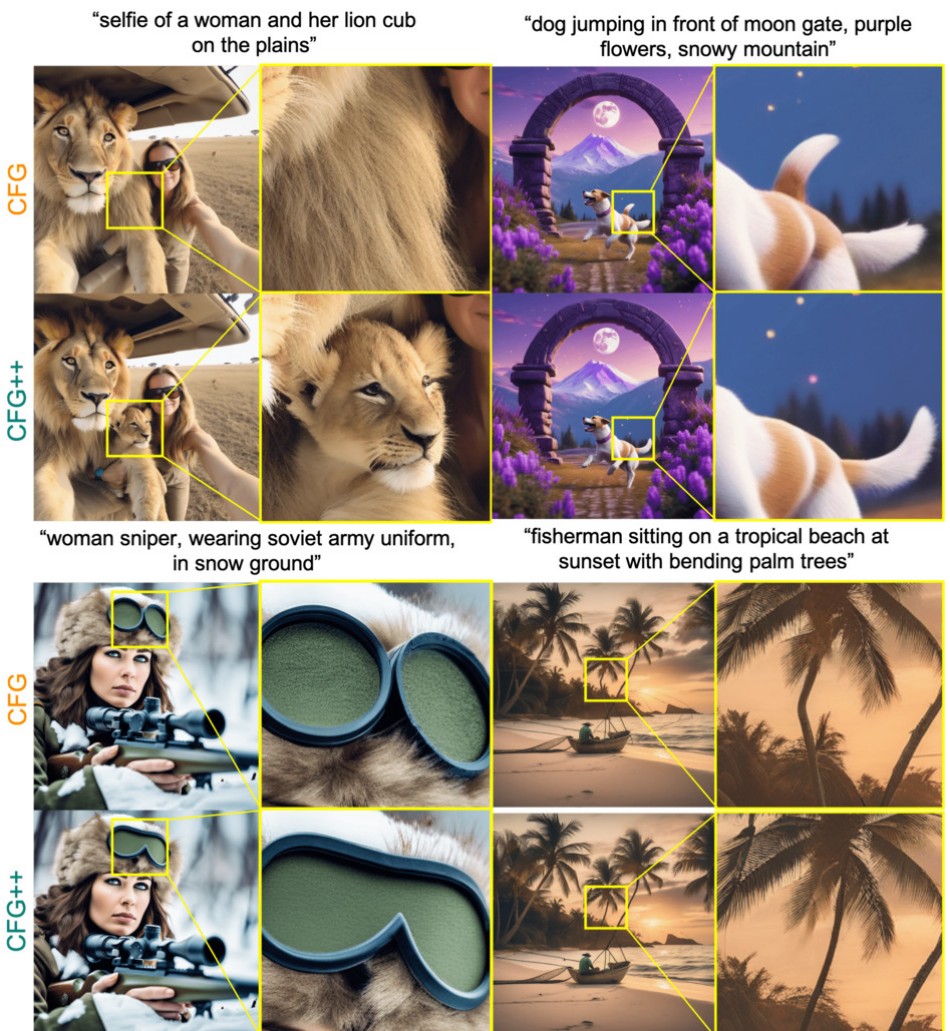

Figure 10: Enhanced T2I results by SDXL ($\omega = 9.0, \lambda = 0.8$) with CFG++. Under CFG, the lion cub is not visible (top-left), the dog appears with two tails (top-right), the goggles have an unusual shape (bottom-left), and the tree trunk is folded (bottom-right). These artifacts are absent in those produced by CFG++.

## F.3 TEXT-CONDITIONED INVERSE PROBLEMS

In Fig. 20-23, we display additional qualitative comparison for text-conditioned inverse problem solver with CFG and CFG++. Experiments are conducted with FFHQ (512x512) validation set and CFG++ consistently leads to better reconstruction of true solution for various tasks.

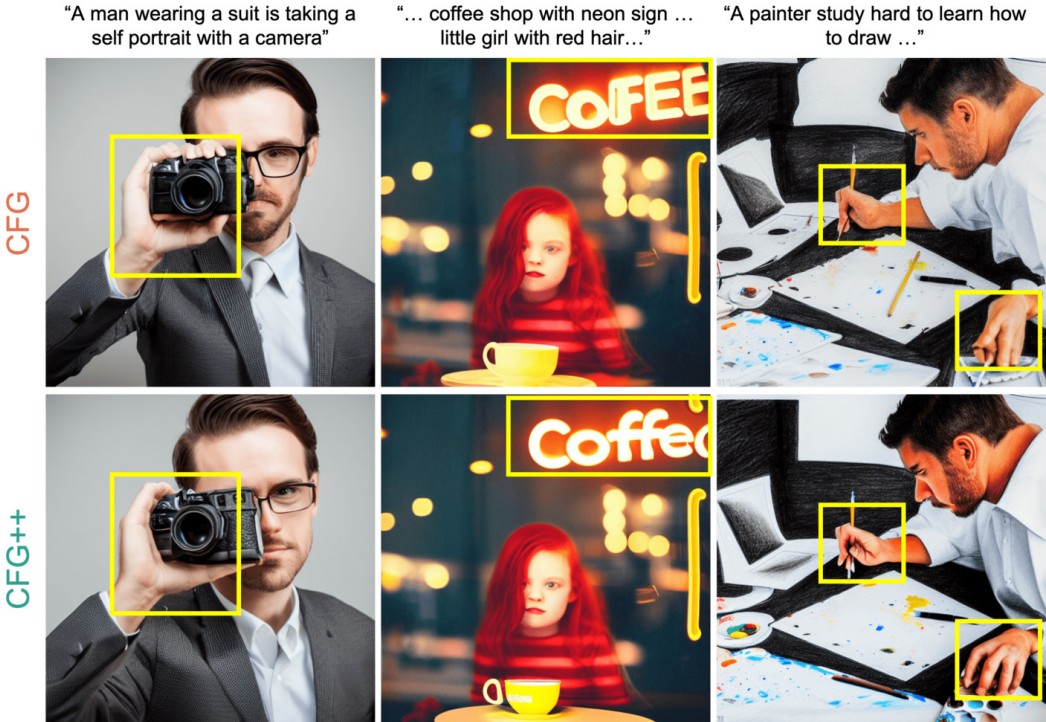

Figure 11: T2I using SD v1.5, CFG vs CFG++ ($\omega = 9.0, \lambda = 0.8$). Unnatural depictions of human hands, and incorrect renderings of the text by CFG are corrected in CFG++.

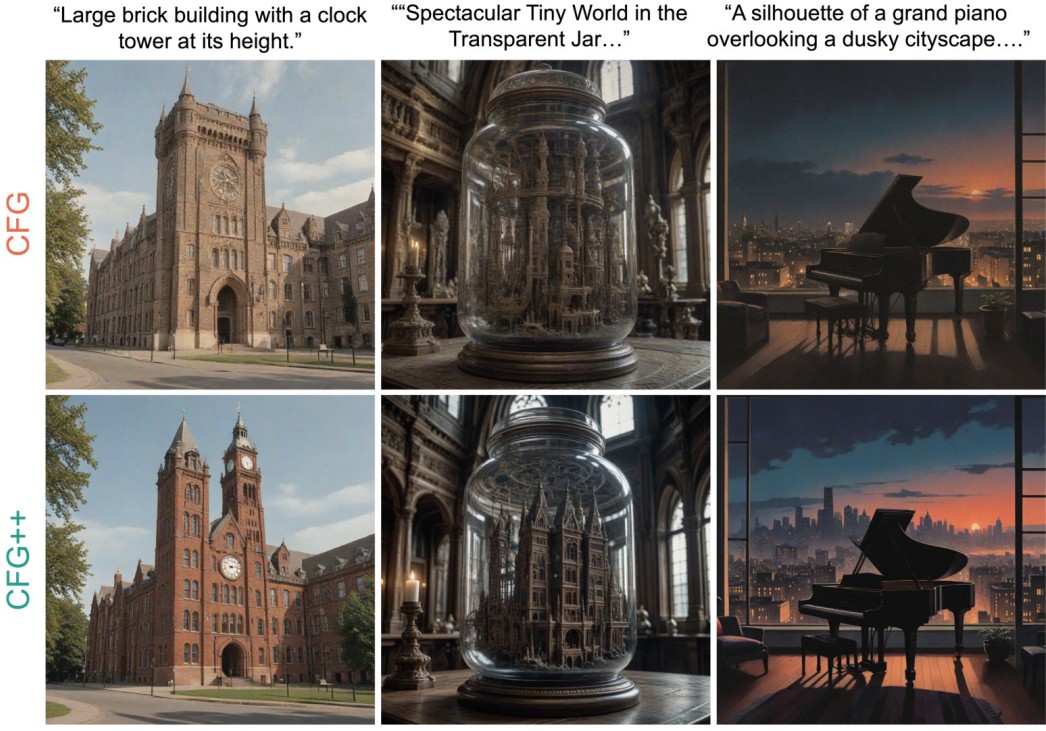

Figure 12: T2I using SDXL-Turbo, 6 NFE, CFG vs CFG++. The overall image quality and sophistication have improved with CFG++. DreamShaper XL was used for both images and metrics in main part.

"An alpaca made of colorful building blocks, cyberpunk." "Floating, colossal, futuristic statue in the sky, …." "A detailed oil painting of an old sea captain, steering his ship…"

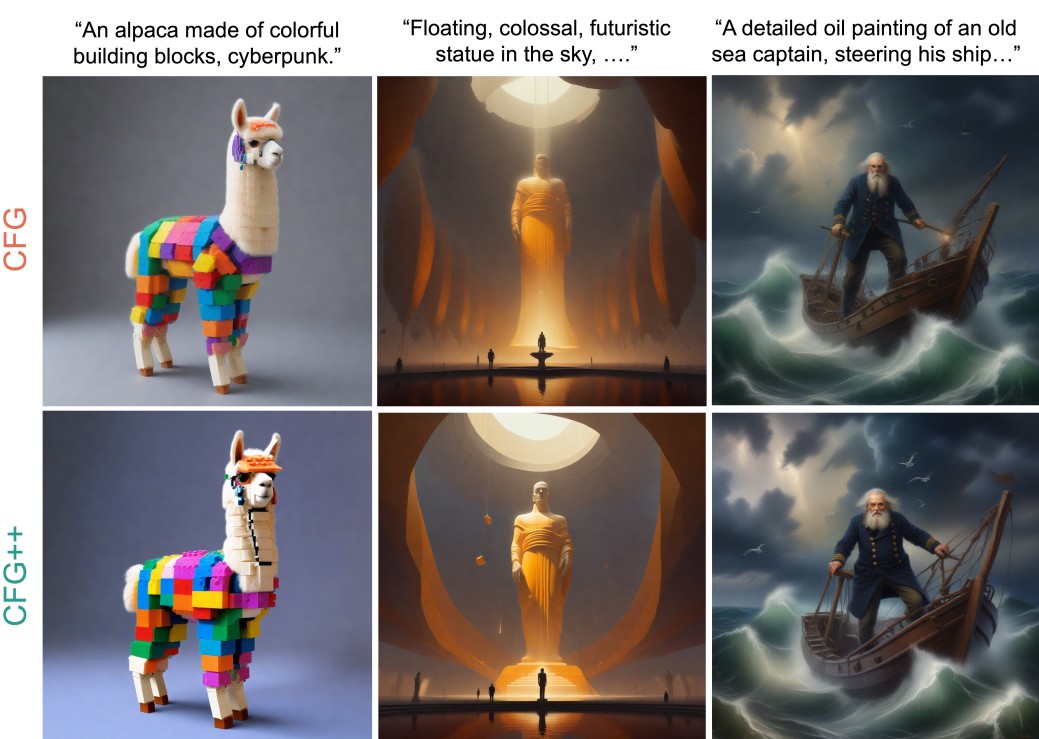

Figure 13: T2I using SDXL-Lightning, 6 NFE, CFG vs CFG++. The overall image quality and sophistication have improved with CFG++. Leosam's HelloWorld XL was used for both images and metrics in main part.

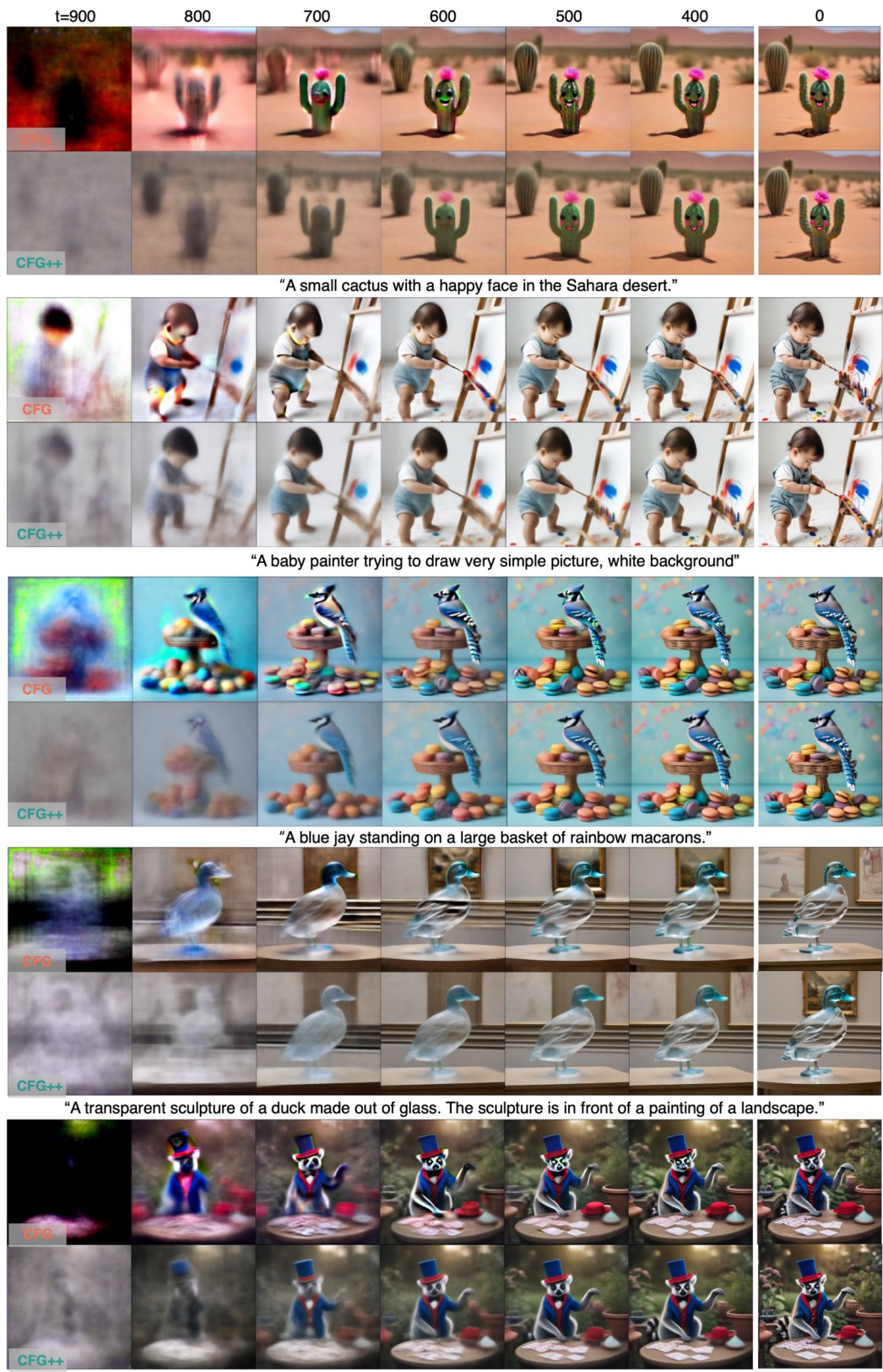

Figure 14: The discrete evolution of the posterior mean in CFG and CFG++. Denoised estimates in latent space are decoded into pixel space at each timestep using SDXL.

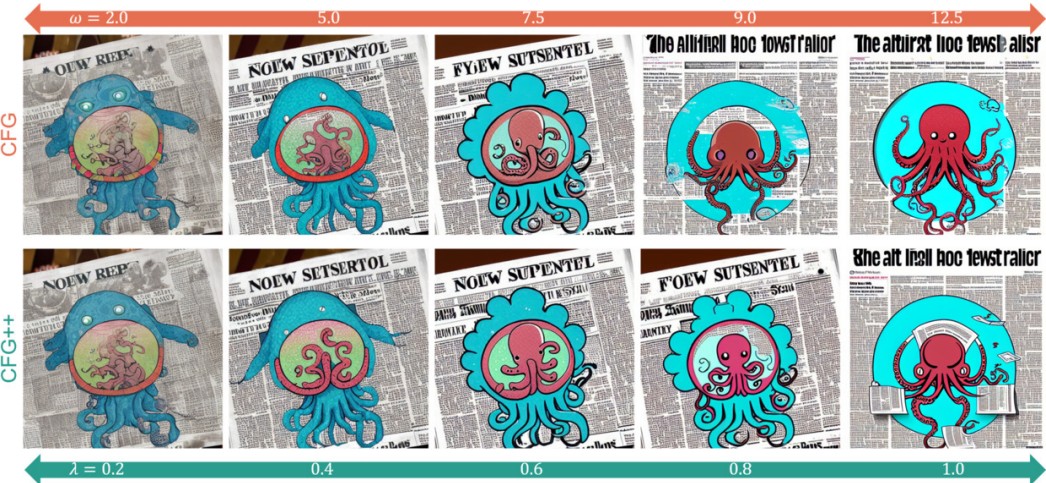

Figure 15: Interpolating behavior of T2I along various guidance scales using SD v1.5 with CFG and CFG++

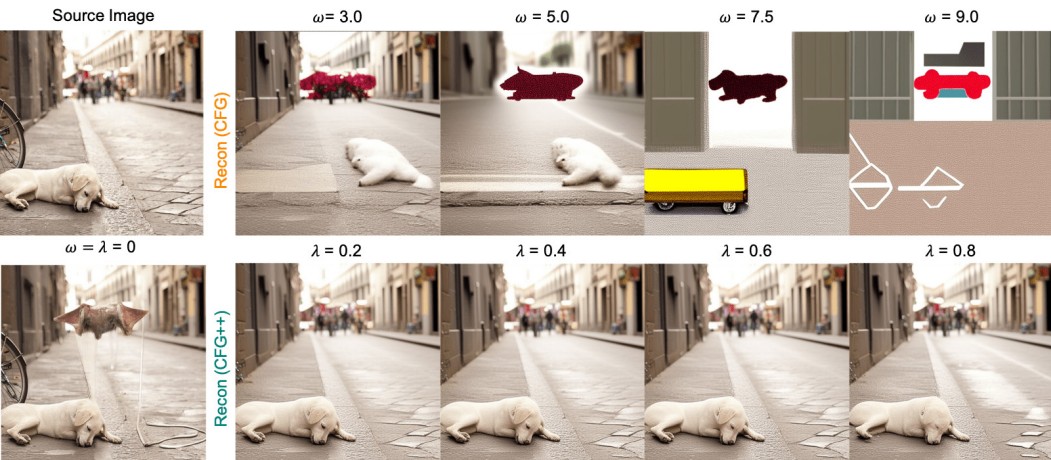

Figure 16: Real image inversion results at various CFG and CFG++ scales using SDv1.5. The image is from COCO dataset. We use the text prompt "a white dog is sleeping on a street and a bicycle" during the inversion.

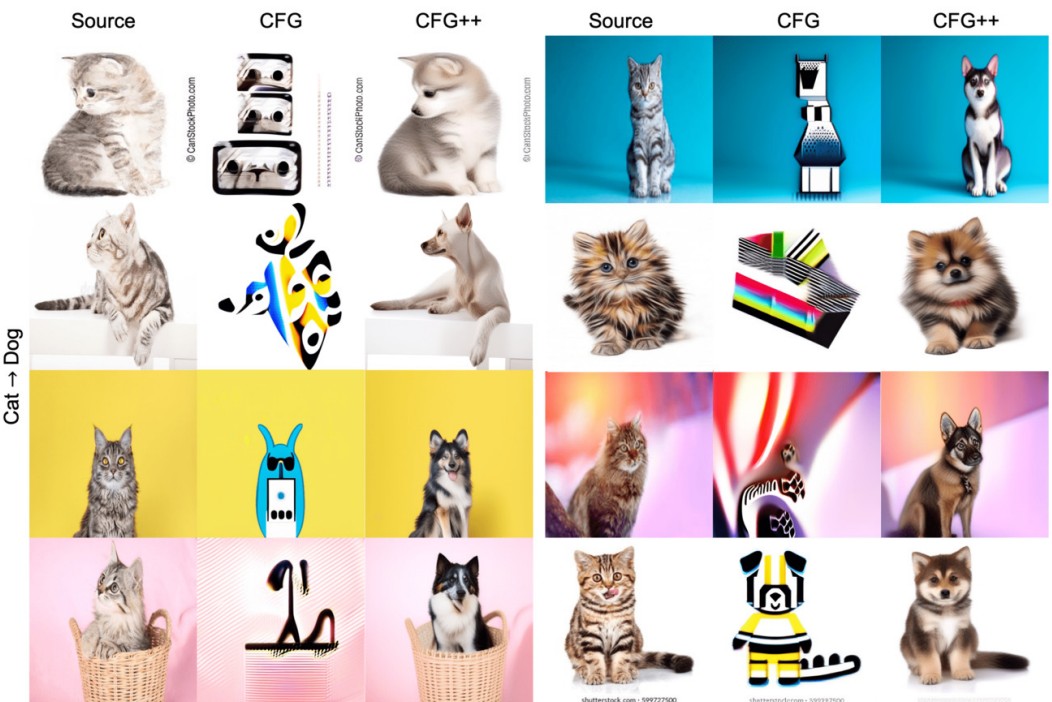

Figure 17: Comparison on real image editing via DDIM inversion under CFG and CFG++ using SDXL. Cat → Dog.

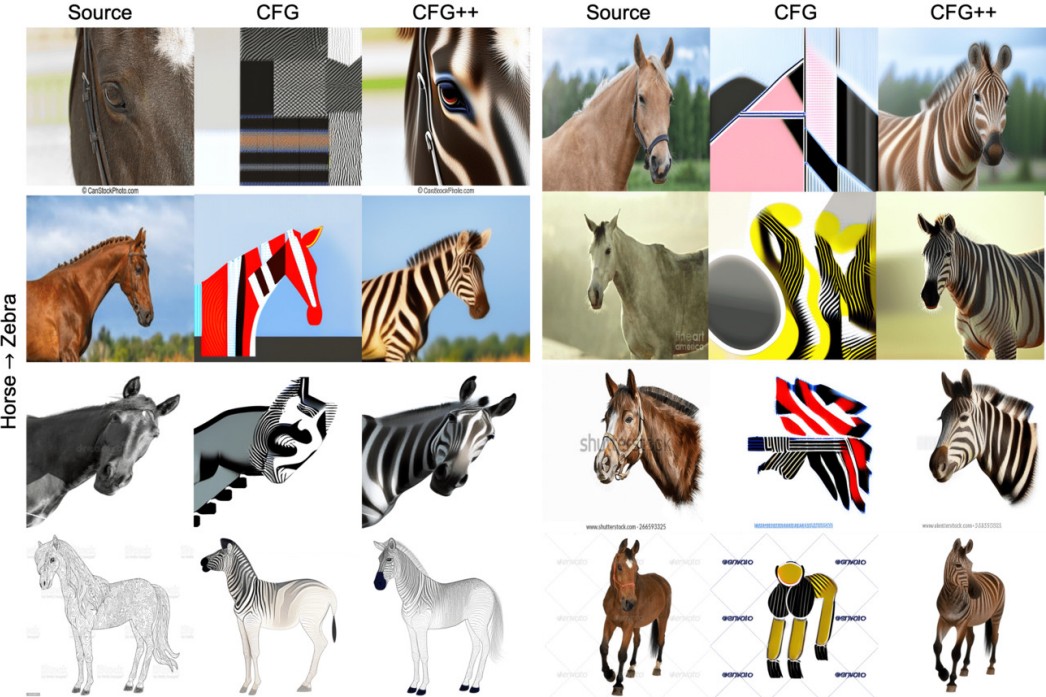

Figure 18: Comparison on real image editing via DDIM inversion under CFG and CFG++ using SDXL. Horse → Zebra.

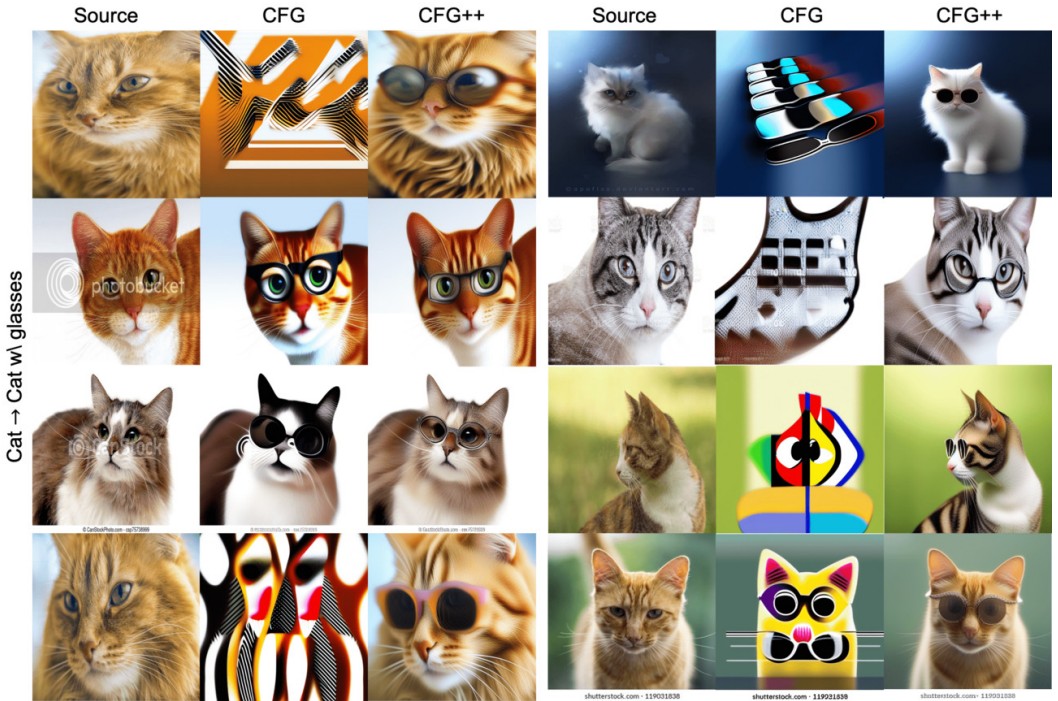

Figure 19: Comparison on real image editing via DDIM inversion under CFG and CFG++ using SDXL. Cat → Cat with glasses.

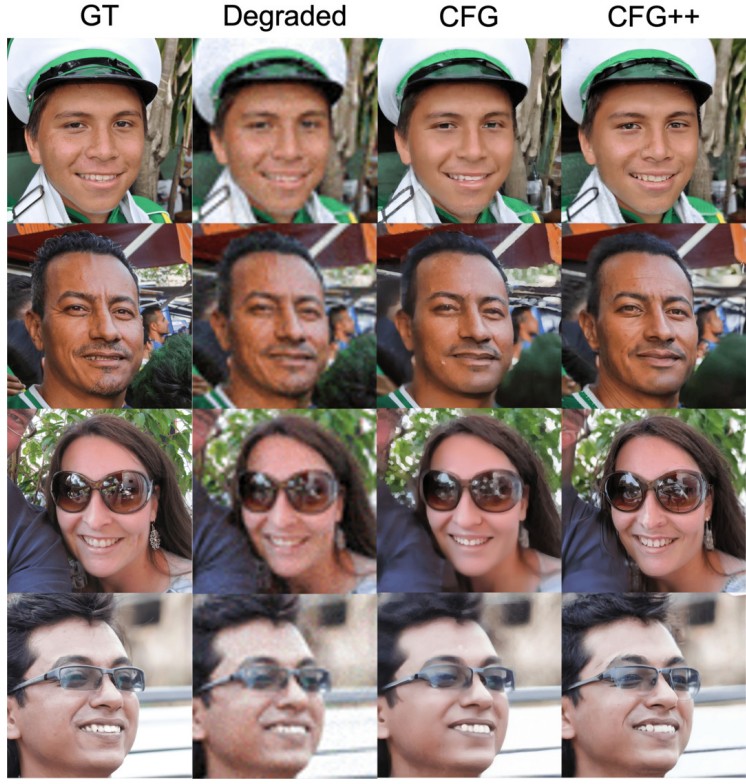

Figure 20: Results of PSLD using CFG and CFG++ on the FFHQ dataset at Super-Resolution (x8).

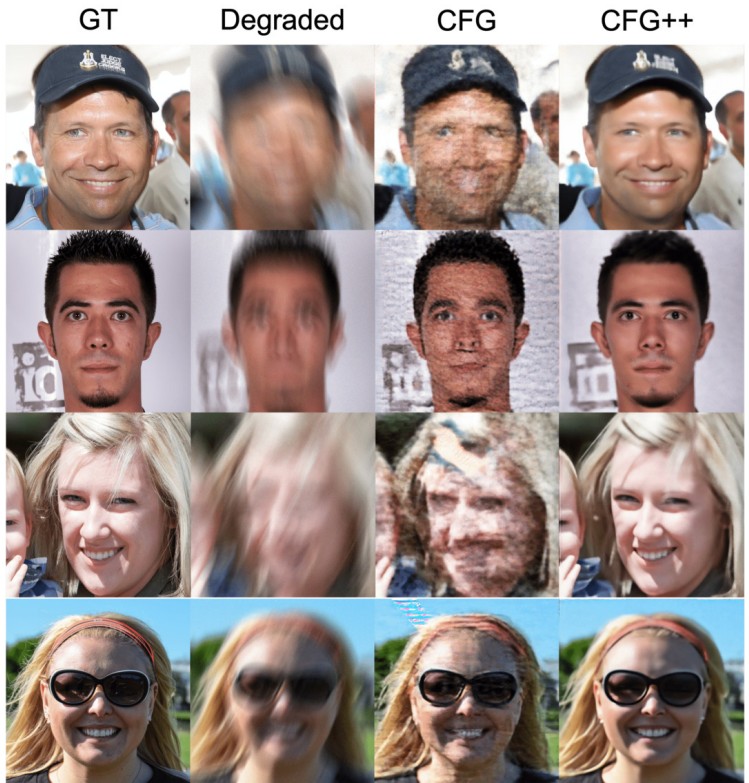

Figure 21: Results of PSLD using CFG and CFG++ on the FFHQ dataset at Motion Deblurring.

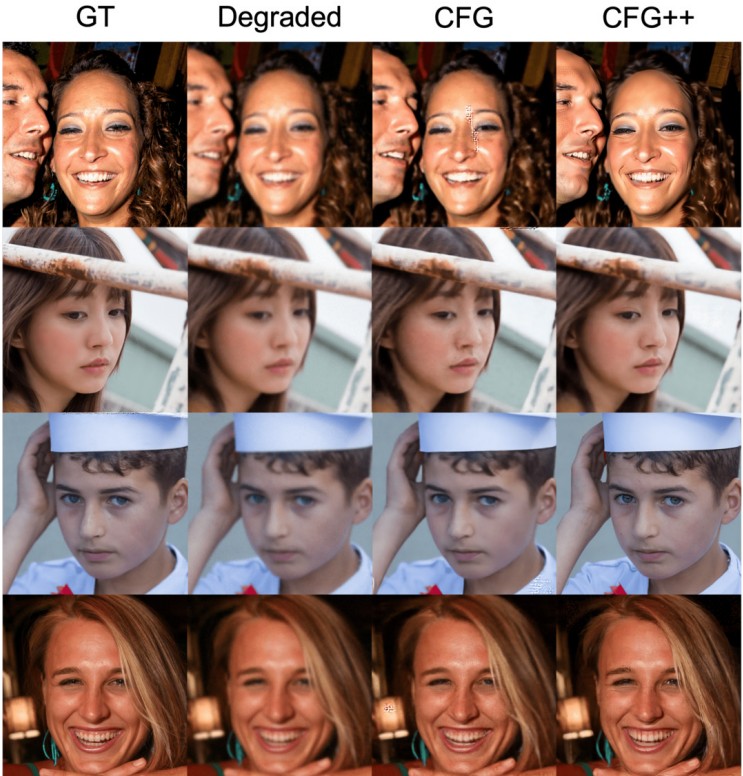

Figure 22: Results of PSLD using CFG and CFG++ on the FFHQ dataset at Gaussian Deblurring.

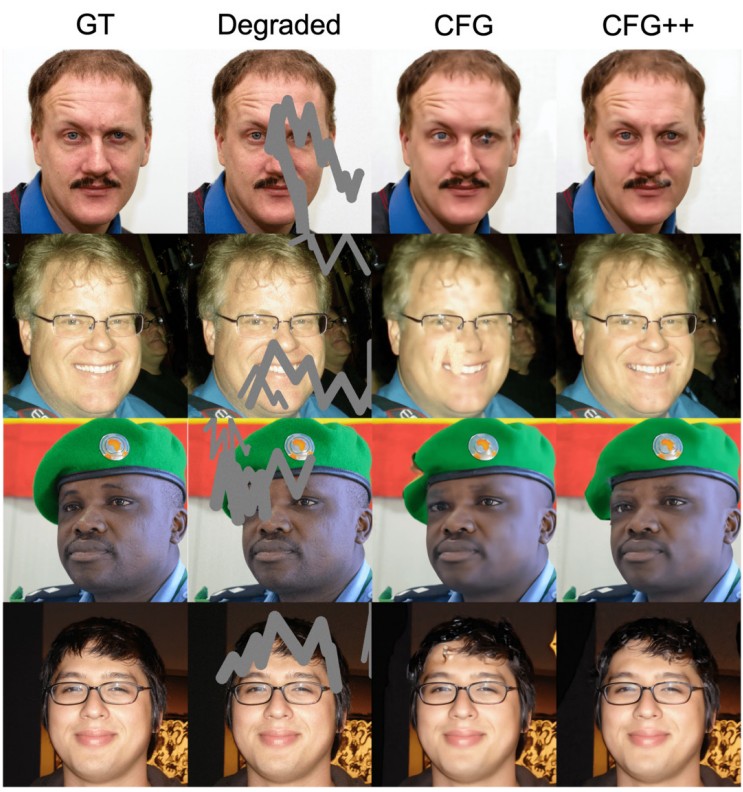

Figure 23: Results of PSLD using CFG and CFG++ on the FFHQ dataset at Inpainting.

