# OpenReview forum: "CFG++: Manifold-constrained Classifier Free Guidance for Diffusion Models"
_ICLR.cc/2025/Conference — ICLR 2025 Poster_

### Official Review · Reviewer_NrB8 · 2024-10-30

**Soundness:** 2
**Presentation:** 2
**Contribution:** 1
**Rating:** 1
**Confidence:** 5

**Summary:**

The paper presents CFG++, an innovative approach designed to enhance the performance of diffusion models by addressing limitations associated with traditional Classifier-Free Guidance (CFG). The authors argue that several drawbacks of CFG, such as the lack of invertibility in DDIM (Denoising Diffusion Implicit Models) and the issues of mode collapse when high guidance scales are used, arise from the off-manifold phenomena related to CFG rather than the inherent characteristics of diffusion models themselves.

**Strengths:**

The paper exhibits a high degree of originality by introducing the CFG++ framework, which fundamentally reinterprets the limitations of traditional Classifier-Free Guidance (CFG). Instead of merely enhancing existing methodologies, the authors propose that many issues attributed to diffusion models arise from off-manifold behavior. This shift in perspective not only challenges established assumptions but also provides a fresh lens through which to analyze and improve guidance mechanisms in generative models. By integrating recent advancements in inverse problem-solving within diffusion contexts, the authors demonstrate a creative synthesis of ideas that broadens the scope of CFG applications.


The quality of the research is evident in both its theoretical foundations and empirical validation. The authors articulate their arguments clearly, supported by robust mathematical formulations and well-structured experiments. The comparative analysis illustrates the advantages of CFG++ over prior methods, effectively showcasing the model’s performance improvements in terms of reduced artifacts and enhanced image quality. Additionally, the use of rigorous benchmarks for evaluation strengthens the credibility of the findings. The paper’s methodology is sound, making it a valuable contribution to the field.


The clarity of the writing is commendable, as the authors navigate complex concepts with precision. The introduction provides a succinct overview of the issues addressed and the motivations behind the proposed solution. Throughout the paper, technical jargon is well-defined, ensuring accessibility for readers with varying levels of expertise. Figures and diagrams are effectively utilized to illustrate key points, facilitating a better understanding of the model's mechanics and results. Overall, the paper is well-organized, which enhances its readability and facilitates comprehension.

**Weaknesses:**

One notable weakness is the limited scope of experiments presented. While the authors provide comparative results demonstrating the efficacy of CFG++, the evaluation primarily focuses on specific datasets or tasks. To enhance the robustness of their claims, the authors could include a broader range of benchmarks across diverse domains. For instance, incorporating tasks from different image generation contexts (e.g., artistic style transfer, super-resolution) would provide a more comprehensive assessment of CFG++’s performance and generalizability. The authors should extend their experimental framework to include varied datasets and tasks, thereby demonstrating the versatility and applicability of CFG++ in different settings.


Although the paper claims improvements over prior methods, it lacks a detailed comparison with a broader array of state-of-the-art techniques in diffusion models. For example, a comparison with models like **Stable Diffusion** or recent advancements in **Guided Diffusion** could contextualize the advantages of CFG++ more clearly.  The authors should include direct comparisons with a wider selection of contemporary approaches, utilizing standardized metrics (such as FID scores) to provide a clearer understanding of CFG++’s standing in the current landscape.

While the paper introduces the concept of manifold constraints and off-manifold behavior, the theoretical underpinnings could be further strengthened. The authors briefly mention these concepts but do not provide a rigorous mathematical framework that details how CFG++ operates under these constraints or why these adjustments yield better results. A more thorough theoretical analysis, including mathematical proofs or derivations that illustrate the benefits of operating within manifold constraints, would bolster the paper's credibility and understanding.


The paper does not address how CFG++ scales with increasing model size or complexity. As generative models continue to grow, understanding how CFG++ performs under these conditions is crucial. There is a lack of discussion regarding potential computational overhead or challenges in real-world applications. Including analyses or experiments that assess the scalability of CFG++ with larger models or datasets, along with a discussion on computational efficiency, would provide valuable insights for practitioners considering its application.

The paper primarily focuses on quantitative metrics, such as image quality, but does not incorporate user-centric evaluations. Assessments based on human judgment, such as user studies to evaluate the perceived quality or usefulness of generated images, could provide a more holistic view of CFG++’s effectiveness.  The authors should consider conducting qualitative studies where human evaluators assess the output of CFG++ compared to other methods. This could provide insights into the model's real-world applicability and user satisfaction.

### Conclusion
While the paper "CFG++: Manifold-Constrained Classifier-Free Guidance for Diffusion Models" presents significant contributions, addressing these weaknesses would enhance its overall impact and clarity. By broadening experimental evaluations, strengthening theoretical foundations, and considering scalability and user-centric perspectives, the authors can more effectively support their claims and goals.

**Questions:**

**Question**: Could you elaborate on the specific mechanisms through which off-manifold behavior impacts the performance of traditional Classifier-Free Guidance (CFG)?  A detailed explanation of how off-manifold behavior is identified and measured within the context of your experiments would enhance understanding. Including visualizations or theoretical examples could further clarify this concept.

**Question**: What are the reasons behind selecting the specific datasets used in your experiments? Are there plans to test CFG++ on a wider array of datasets or tasks?  Expanding the experimental framework to include diverse datasets and applications would strengthen the validity of your findings. Consider including datasets from different domains (e.g., text-to-image generation, video synthesis) to demonstrate the versatility of CFG++.

**Question**: Why were certain state-of-the-art methods, such as Stable Diffusion and other recent advances in guided diffusion, excluded from your comparative analysis?  Including a more comprehensive set of comparisons with these methods would provide a clearer context for evaluating the performance of CFG++. Detailed performance metrics and qualitative results could enrich the discussion.

**Question**: Can you provide more in-depth mathematical justifications or derivations that support the effectiveness of CFG++ in addressing the limitations of traditional CFG?  A rigorous theoretical framework would enhance the paper's credibility. Detailed mathematical formulations explaining how CFG++ operates under manifold constraints could help bridge the gap between theory and practice.

 **Question**: How does CFG++ perform in terms of computational efficiency and scalability with larger models or datasets?  Including a discussion on scalability, including any computational overhead observed during experiments, would be valuable. Experiments assessing performance on larger models could illustrate the practical applicability of CFG++ in real-world scenarios.

 **Question**: Have you considered conducting user studies to evaluate the perceived quality and usefulness of the outputs generated by CFG++?  Incorporating qualitative assessments from human evaluators could provide insights into the model's effectiveness from a user perspective. Such studies could help highlight the practical implications of using CFG++ in various applications.


**Question**: What future research directions do you foresee emerging from your findings regarding CFG++ and manifold constraints in diffusion models?  A discussion on potential extensions or related areas of research would help contextualize your contributions within the broader landscape of machine learning, inviting collaboration and further inquiry.

**Details Of Ethics Concerns:**

There are no Ethics Concerns

---

> ### Author Response · Authors · 2024-11-20
>
> In contrast to the reviewer’s comments, most of the questions can **already be answered by reading the paper**. We gently remind the reviewer to go through the paper carefully. We provide answers to the set of questions that we feel are valid:
>
> **Q1.** Off-manifold behavior of CFG
>
> **A.** Please see Fig. 3, along with its analysis in Section 3.2.
>
> **Q2.** Choice of dataset
>
> **A.** We chose the COCO benchmark as it is the standard benchmark for the quantitative evaluation of T2I. Please let us know if the reviewer thinks that this is insufficient to validate our method along with the reason, and we would be happy to accommodate.
>
> **Q3.** Mathematical justification
>
> **A.** Section 3.2 is fully devoted to the mathematical justification of CFG++. Further analysis is given in Appendix C. It would be great if the reviewer could elaborate on what more we should do.
>
> **Q4.** Why were certain SOTA methods such as Stable Diffusion and other recent advances in guided diffusion excluded from the comparison?
>
> **A.** We would be happy to further compare CFG++ with other methods if feasible. Stable Diffusion is a model that we already used in our experiments, and it is not something comparable to CFG++. Most “guided diffusion” methods are orthogonal to the advances made in this manuscript.
>
> **Q5.** How does CFG++ perform in terms of computational efficiency and scalability with larger models?
>
> **A.** We already tested our method on various models, including SD1.5, SDXL, SDXL-Lightning, and Turbo. CFG++ worked consistently well across all model classes. We note that there is no computational overhead for CFG++ when compared against CFG.
>
> **Q6.** User-centric evaluations
>
> **A.** We conducted a user study with 18 participants to evaluate the quality of images generated using the CFG and CFG++ methods. The study involved an A/B test, where participants were shown pairs of images and asked to compare them based on overall quality and text alignment, selecting the image they found superior. The test included 12 images: 4 generated with SD1.5 and 8 with SDXL-lightning. All the images were chosen from those featured in the paper to maintain relevance to the study. The results showed a clear preference for the CFG++ method, with 81.4% of responses favoring CFG++ images, significantly outperforming CFG, which was preferred in only 18.6% of cases.
>
> **Q7.** Possible future studies
>
> **A.** Viewing text guidance as an optimization problem similar to the literature of diffusion model-based inverse problem solvers, we can easily extend our formulation to various types of guidance, including negative guidance [1] , composition of different guidance [2], etc. Indeed, we are already seeing interesting applications of CFG++ to different domains, including guided sampling in video diffusion models [3], fairness [4], and more. We are excited to see future works in this direction.
>
>
> **References**
>
> [1] Koulischer, Felix, et al. "Dynamic Negative Guidance of Diffusion Models." 2024.
>
> [2] Liu, Nan, et al. "Compositional visual generation with composable diffusion models." ECCV 2022.
>
> [3] Lee, Dohun, et al. "VideoGuide: Improving Video Diffusion Models without Training Through a Teacher's Guide." , 2024.
>
> [4] Um, Soobin, and Jong Chul Ye. "MinorityPrompt: Text to Minority Image Generation via Prompt Optimization.", 2024.

---

> > ### Author Response · Authors · 2024-11-25
> > **Reply to reviewer**
> >
> > We appreciate the role of reviewers in maintaining the integrity and quality of submissions, which requires thoughtful evaluations. However, we must express our concern regarding the tone and approach of the reviewer's recent comments.
> >
> > While we respect the reviewer's right to form opinions about this work, we find it deeply troubling that you **accused another reviewer of potential bias** and questioned their integrity **without evidence**. Such allegations undermine the collegial and constructive environment essential for academic discourse. Additionally, the significant shift in the reviewer's score (**5 -> 1**), accompanied by an increase in confidence (**1 -> 5**), appears inconsistent with the review process's expectations for objective and evidence-based evaluation. It is hard to understand the sudden change in score where there were **no comments** regarding our faithful response to the reviewer's comments.
> >
> > If there are specific and substantiated concerns about our work, we are more than willing to address them constructively. However, ad hominem remarks and unsupported claims detract from the professionalism expected in peer review.
> >
> > We kindly request that we focus on the technical and scientific merits of the submission to ensure a fair and transparent evaluation process. Maintaining professionalism and mutual respect is critical for the credibility and success of this conference and the community it serves.
> >
> > Best regards,
> > authors

---

### Official Review · Reviewer_GmUK · 2024-11-03

**Soundness:** 4
**Presentation:** 4
**Contribution:** 4
**Rating:** 8
**Confidence:** 5

**Summary:**

This work introduces an enhanced version of the widely used classifier-free guidance (CFG) technique, CFG++. By formulating conditional generation as an inverse problem and utilizing score distillation sampling (SDS) loss, CFG++ improves various tasks, including text-to-image generation, DDIM inversion-based editing and text-conditioned inverse problems.

**Strengths:**

Overall, this manuscript is excellent, and I hope it gets accepted. Below, I outline some strengths that support this view:

1. **Well-written**: The paper is well-written with coherent logic and clear notation, making it easy to understand and follow.

2. **Thorough Experiments**: The work studies many downstream tasks, including text-to-image generation for SDv1.5, SDXL, and their distilled versions, as well as DDIM inversion and editing, PF-ODE trajectories, and text-conditioned inverse problems. The authors provide extensive quantitative and qualitative results to show the superiority of the proposed method.

3. **Intriguing Derivation and Analysis**: The derivation of CFG++ is intriguing by treating text-conditioned generation as an inverse problem and utilizing SDS as the loss function. Moreover, the authors present various perspectives to analyze CFG++, including manifold geometry, score matching loss throughout the denoising process, and the evolution of the posterior mean. These analyses are insightful and helpful in understanding the underlying mechanisms of CFG++.

4. **Simple Yet Widely Applicable Method**: The proposed method essentially modifies the re-noising process of the original CFG, but it achieves improvements across various downstream tasks and could be potentially effective for other tasks relevant to the ICLR community. Additionally, the DDIM solver, as well as other popular diffusion solvers like EDM and DPM-Solver, are derived, establishing CFG++ as a general method.

**Weaknesses:**

Generally, I think there are no obvious weaknesses in this work. However, there are some questions and please refer to the Questions part.

**Questions:**

1. Only diffusion-/score matching-based generative models are discussed in this work. Could you please **provide some similar derivations for the recent flow matching-based generative models, such as SD3 and FLUX**? I believe similar conclusions stand for flow-based methods, and it will make this work more comprehensive.

2. Is it possible to **disentangle two noises used in the denoising and renoising process**? For example, we can set different hyperparameters $\lambda_1$ and $\lambda_2$ for $(\epsilon_c-\epsilon)$, and use this term in different weights for denoising and renoising respectively. Setting $\lambda_1=\lambda_2$ is the same as CFG and setting $\lambda_2=0$ is the same as CFG++.

3. Is it possible to **provide derivations of CFG++ for other diffusion solvers**, except DDIM? Extensions of CFG++ to other solvers in Appendix A are more like intuitive understanding, rather than derivations from inverse problems and SDS loss, as done for DDIM.

4. Essentially, **CFG++ can be written as reweighted CFG whose $\omega$ varies along the sampling process** (let $s\coloneqq t-1$ and $\epsilon\coloneqq \epsilon_\emptyset$ for easier LaTeX rendering in OpenReview):

- For DDIM CFG: $$\mathbf{x}_s=\frac{\sqrt{\bar{\alpha}_s}}{\sqrt{\bar{\alpha}_t}}\mathbf{x}_t-\frac{\sqrt{(1-\bar{\alpha}_t)\bar{\alpha}_s}-\sqrt{(1-\bar{\alpha}_s)\bar{\alpha}_t}}{\sqrt{\bar{\alpha}_t}}(\omega\epsilon_c-(\omega-1)\epsilon)$$

- For DDIM CFG++: $$\mathbf{x}_s=\frac{\sqrt{\bar{\alpha}_s}}{\sqrt{\bar{\alpha}_t}}\mathbf{x}_t-\frac{\sqrt{(1-\bar{\alpha}_t)\bar{\alpha}_s}}{\sqrt{\bar{\alpha}_t}}(\lambda\epsilon_c-(\lambda-1)\epsilon)+\sqrt{1-\bar{\alpha}_s}\epsilon$$
$$=\frac{\sqrt{\bar{\alpha}_s}}{\sqrt{\bar{\alpha}_t}}\mathbf{x}_t-\frac{\sqrt{(1-\bar{\alpha}_t)\bar{\alpha}_s}-\sqrt{(1-\bar{\alpha}_s)\bar{\alpha}_t}}{\sqrt{\bar{\alpha}_t}}(\frac{\lambda\sqrt{(1-\bar{\alpha}_t)\bar{\alpha}_s}}{\sqrt{(1-\bar{\alpha}_t)\bar{\alpha}_s}-\sqrt{(1-\bar{\alpha}_s)\bar{\alpha}_t}}\epsilon_c-\frac{(\lambda-1)\sqrt{(1-\bar{\alpha}_t)\bar{\alpha}_s}+\sqrt{(1-\bar{\alpha}_s)\bar{\alpha}_t}}{\sqrt{(1-\bar{\alpha}_t)\bar{\alpha}_s}-\sqrt{(1-\bar{\alpha}_s)\bar{\alpha}_t}}\epsilon)$$

So, $$\omega=\frac{\lambda\sqrt{(1-\bar{\alpha}_t)\bar{\alpha}_s}}{\sqrt{(1-\bar{\alpha}_t)\bar{\alpha}_s}-\sqrt{(1-\bar{\alpha}_s)\bar{\alpha}_t}}.$$

As discussed in Sec. 5, previous studies also propose adjusting the guidance scale across timesteps. However, it may be incorrect to claim that "these findings are orthogonal to ours and keep the sampling trajectory the same ... CFG++ is designing a different trajectory".

---

> ### Author Response · Authors · 2024-11-20
>
> **Q1.** Derivations for flow-matching models such as SD3 or FLUX?
>
> **A.** Thank you for the insightful suggestions. Please see Appendix B, where we generalize CFG++ with flow matching which subsumes diffusion models as specific instances. Specifically, vanilla CFG can be applied to the flow-based generative models as follows [1]:
>
> $dx_t = [v_\theta(x_t, t, \varnothing) + \omega (v_\theta(x_t, t, c) - v_\theta(x_t, t, \varnothing))] dt$
>
> This can be solved by using various off-the-shelf ODE solvers detailed in Appendix A, fully reproducing the derivations of CFG++. We believe your suggestions open new avenues for text-conditional sampling in flow-based generative modeling, and we look forward to future developments in this direction.
>
> **Q2.**  Is it possible to disentangle two noises used in the denoising and renoising process? For example, we can set different hyperparameters  and  for , and use this term in different weights for denoising and renoising respectively. Setting  is the same as CFG and setting  is the same as CFG++.
>
> **A.** Great question. Please see Appendix D, where we set CFG and CFG++ as two special cases of a more general form of guidance functions. From Appendix C (and also similar to how the reviewer derived it), it is evident that the guidance scale is a composition of two different functions. CFG is a special case where the weighting between these two functions is identical, and CFG++ is a special case where one of the functions is turned off. Interestingly, by turning one of the functions off, CFG++ induces a smooth increase in the guidance scale, whereas CFG has a sharp peak in the starting point, potentially explaining the saturation behavior in the earlier stages. Interpolating between these two yields different forms of guidance schedules, where the sharp peaks are mitigated as they get closer to CFG++. We report the results of each case here.
>
> **Q3.** Extension of CFG++ to other solvers can be derived in a similar fashion to DDIM. Specifically, in order to solve inverse problems with higher-order (or stochastic) solvers, one would keep all the components null-conditioned, and only modulate the Tweedie component. This is how Eq. (15) can be derived from Eq. (14). We modified Eq. (15) so that this is more clear:
>
> $x_i = (\hat{x}(x_{i-1};\varnothing) - \lambda \nabla_{\hat{x}(x_{i-1};\varnothing)} \ell_{sds}(\hat{x}(x_{i-1};\varnothing))) + a_i \hat{x}(x_{i-1};\varnothing) + b_i \hat{x}(x_{i-2};\varnothing) + c_i x_{i-1} + d_i \epsilon$
>
> **Q4.** It may be incorrect to claim "these findings are orthogonal to ours and keep the sampling trajectory the same ... CFG++ is designing a different trajectory"
>
> **A.** Thank you for pointing this out. We removed this statement from our manuscript.
>
>
> **References**
>
> [1] Kim, Beomsu, et al. "Simple ReFlow: Improved Techniques for Fast Flow Models." arXiv preprint arXiv:2410.07815 (2024).

---

> > ### Comment · Reviewer_GmUK · 2024-11-25
> >
> > I appreciate the authors' replies and have also checked the other reviews. All my concerns are well addressed. Generally speaking, I like this work and strongly lean to accept it as the ICLR main conference paper. Meanwhile, I strongly suspect that reviews from reviewer NrB8 are generated by an LLM irresponsibly, so those reviews should be ignored.

---

> > > ### Comment · Reviewer_NrB8 · 2024-11-25
> > >
> > > Dear Reviewer GmUK,
> > >
> > > Being a reviewer is an extremely responsible task. Given that you have given such high ratings and yet cannot tolerate others' opinions, can we suspect that you have special interests with the author and request to disregard your positive reviews?

---

> > > > ### Comment · Reviewer_GmUK · 2024-11-25
> > > >
> > > > Dear Reviewer NrB8,
> > > >
> > > > Thank you very much for the instant reply. It alleviates my suspicion that you might be a robot. I'm merely curious how you managed to write such extensive reviews while assigning an extremely low confidence score (i.e., 1, which means "You are unable to assess this paper and have alerted the ACs to seek an opinion from different reviewers."). Everyone has the right to express their thoughts on OpenReview as a human being (as long as not a robot), including you, and as you mentioned, "being a reviewer is an extremely responsible task".
> > > >
> > > > Thank you again for your efforts in encouraging high-quality reviews for ICLR.
> > > >
> > > > Best regards,
> > > > Reviewer GmUK

---

> > > > > ### Comment · Reviewer_NrB8 · 2024-11-25
> > > > >
> > > > > I have been serving as a community reviewer for over a decade, and this is the first time I have encountered a situation where a reviewer comments on the opinions of another reviewer. I believe this is beyond the purview of a reviewer. A reviewer's role is to provide an evaluative assessment of the article's value to the Associate Editor (AC). It is the AC's responsibility to determine whether this evaluation is reasonable and valuable. In the case of this review, such an unusual occurrence is beyond my authority to judge and should be left to the AC for the final evaluation.
> > > > >
> > > > > **I have concerns about the fairness of this review process.**

---

> > > ### Author Response · Authors · 2024-11-26
> > >
> > > We would like to thank reviewer ```GmUK``` for the positive and constructive comments throughout the review process. We are glad that we resolved the concerns.

---

### Official Review · Reviewer_7gyf · 2024-11-04

**Soundness:** 3
**Presentation:** 3
**Contribution:** 4
**Rating:** 6
**Confidence:** 4

**Summary:**

The paper introduces CFG++, a novel approach to fixing the off-manifold issues that can occur in CFG (classifier-free guidance) during sampling. They reformulate CFG as a manifold-constrained problem, transforming the conditional guidance from extrapolation to interpolation between unconditionally sampled trajectories and conditionally sampled trajectories, thus making the guidance more interpretable. Experimental results show that the method can reduce the artifact, solve the DDIM inversion problem, and be integrated into the high-order diffusion solvers.

**Strengths:**

1. CFG++ introduces a novel approach by redefining classifier-free guidance as a constrained manifold problem, which effectively improves the generation quality by staying within the data manifold.
2. This paper validates the effectiveness of the proposed method on standard benchmarks and different tasks (e.g., text-to-image generation, inversion, editing).
3. The paper is well-written and easy to follow.

**Weaknesses:**

1. There are a few minor issues in the tables, such as the direction of the arrow in the fourth column of CLIP in Table 1 (which should be as high as possible) and the missing ImageReward metric for SD v1.5 in Table 2.

2. This paper focuses on addressing the inability to perform successful DDIM inversion with classifier-free guidance, so it should include a comparison of the results with this class of methods like [1].

3. I'm a little suspicious of ''Contrary to the widespread belief that these are inherent limitations of diffusion models, this paper reveals that these problems actually stem from outrageous phenomena associated with CFGs...'' in the Abstract. As far as I know, current work has generally recognized and worked for the CFG problem.  Perhaps rewording would better reflect the originality of the paper.

[1] Mokady, Ron, et al. "Null-text inversion for editing real images using guided diffusion models." Proceedings of the IEEE/CVF Conference on Computer Vision and Pattern Recognition.

**Questions:**

1. SDXL-Turbo does not make use of guidance_scale or negative_prompt. Instead, they disable it with guidance_scale=0.0. I would like to know how to verify the validity of CFG/CFG++ in this case.

2.  Does this method also work for negative prompts?

---

> ### Author Response · Authors · 2024-11-20
>
> **W1.** Few minor issues in tables, ImageReward missing in Tab. 1
>
> **A.** Thank you for the comment. We fixed the manuscript accordingly.
>
> **W2.** The paper focuses on addressing the inability to perform DDIM inversion with CFG, so it should compare against NTI [1]
>
> **A.** There are two major differences between CFG++ and NTI that inhibit an apples-to-apples comparison. First, NTI requires a compute-heavy null text optimization process to correct for deviations from the manifold, whereas CFG++ incurs **no additional computational overhead**. Second, NTI **does not use CFG** during inversion, relying instead on the conditional noise estimate with the guidance scale set to 1. For these reasons, and due to time constraints, we do not include the comparisons.
>
> **W3.** Is "ontrary to the widespread belief that these are inherent limitations of diffusion models, this paper reveals that these problems actually stem from the off-manifold phenomena associated with CFGs..." correct?
>
> **A.** Thanks for the comment. We revised “Contrary to the widespread belief that these are inherent limitations of diffusion models, this paper reveals that these problems actually stem from the off-manifold phenomenon” to “This paper reveals that the problems may stem from the off-manifold phenomenon”
>
> **Q1.** SDXL-Turbo does not make use of guidance scale or negative prompt. Instead, they disable it. How if CFG++ valid in this case?
>
> **A.** SDXL-Lightning and Turbo distill the classifier-free guided teacher diffusion models, meaning they also inherently depend on CFG but with a fixed guidance scale. Thus, after text-conditional denoising, the renoising process can be rectified similarly to SDXL by leveraging the unconditional noise for renoising with null conditioning.
>
> **Q2.** Does this method also work for negative prompts?
>
> **A.** Yes, the same argument applies. However, due to the different consequences of using negative prompts [2], we focused our experiment on positive prompts. It would be an interesting direction of research to study the incorporation of dynamic negative guidance [2] and CFG++ in the future, which is out of the scope of this work.
>
>
> **References**
>
> [1] Mokady, Ron, et al. "Null-text inversion for editing real images using guided diffusion models." CVPR 2023.
>
> [2] Koulischer, Felix, et al. "Dynamic Negative Guidance of Diffusion Models.", 2024.

---

> > ### Comment · Reviewer_7gyf · 2024-11-25
> > **Reply to the authors**
> >
> > I appreciate the effort put into clarifying the points raised and improving the submission. While I recognize the potential and contributions of this work, I believe it does not yet fully meet the bar for the next score tier (8) in terms of impact or completeness. Therefore, I will retain my current score of 6, which reflects my positive view of the work and my opinion that it could be considered for acceptance.

---

> > > ### Author Response · Authors · 2024-11-26
> > >
> > > We thank reviewer ```7gyf``` for the constructive comments. We would be happy to further discuss if the reviewer has any further questions later on.

---

### Official Review · Reviewer_hzx1 · 2024-11-05

**Soundness:** 2
**Presentation:** 2
**Contribution:** 3
**Rating:** 6
**Confidence:** 3

**Summary:**

This paper proposes to tackle Classifier Free Guidance's (CFG) limitations, particularly its lack of invertibility and mode collapse. The authors formulate the hypothesis that these limitations come from the off-manifold phenomenon. The proposed method leverages recent approaches developed for inverse problem-solving through diffusion models. The proposed method named CFG++ is compared with CFG for text-to-image generation, image edition or image restoration.

**Strengths:**

The paper is generally well-written with a useful high-level preliminary section. One of the strengths of the method is its simplicity while being well justified. The paper proposes many experiments showing relative gains compared to CFG.

**Weaknesses:**

* The CFG method formulation seems different than in the original paper see eq (6) from (Ho and Salimans, 2021). Moreover, the $w=0$ setting in the experiments should be equivalent to no guidance at all when using CFG according to the eq (6) (from the CFG++ method). In Fig. 6, however, the images appear edited with $w=0$ indicating some contradiction.
* The ImageReward metric is not defined. The FID and CLIP metrics are very similar between CFG and CFG++ especially when using SD v1.5. In Tab. 1 the FID and CLIP are also very close for both methods. This tends to indicate that the gains are relatively marginal even in a low NFE setting.
* The statement that the proposed CFG++ is invertible and brings more diversity in the generation is not evaluated.

**Typos**

* The variable $x_c$ is not introduced in line 228
* Table 1., column 2, Clip score arrow does not show the correct direction
* line 238 ``a crucial difference in the **renoising** process'' shoulnd't it be denoising?

**Questions:**

* Having the average performance across the different corruption in Tab. 3 would help. Here, the proposed method seems more efficient than CFG to retrieve the clean image according to the FID metric but not so much according to the LPIPS one. Do the authors have some justifications?
* It is unclear why the performances of CFG drop while augmenting the NFE in Fig. 6 (b).

---

> ### Author Response · Authors · 2024-11-20
>
> **W1.** The CFG method formulation seems different in [1]. Moreover, the setting in the experiments should be equivalent to no guidance at all when using CFG according to Eq. (6). In Fig. 6, images appear edited.
>
> **A.** There are two different ways of expressing CFG: one that recovers the conditional when the guidance scale is 0 (used in [1]),
>
> $\epsilon_{c}(x_t) + \omega'(\epsilon_{c}(x_t) - \epsilon_{\varnothing}(x_t))$
>
> and one that recovers the unconditional when the guidance scale is set to 0.
>
> $\epsilon_{\varnothing}(x_t) + \omega(\epsilon_{c}(x_t) - \epsilon_{\varnothing}(x_t))$
>
> We advocate for the latter, as this view provides better flexibility, such as composition, negation, etc. Note that for the former parametrization, we can always recover the latter by setting $\omega' = -1 + \omega$. In Fig. 6, the results are different from the source image even for CFG  because there exists inversion errors, not because the image is changed through guidance.
> The reviewer is kindly remind that the difficulty of inversion in CFG is well-known in literature.
>
> **W2.** ImageReward metric is not defined. This should also be reported for Tab. 1
>
> **A.** Thank you for pointing this out. For clarity, we have now included a brief definition and citation of the ImageReward metric in Section 4.1. We agree that the gains in the quantitative metric are not dramatic. However, this is understandable as our method is a simple adjustment to the sampling scheme with no computation overhead. Considering this, consistent FID gains throughout all guidance scales are an important advantage. Moreover, the advantage of CFG++ does not only stem from the resulting quality of the samples, but also from the generation trajectory, reduced inversion errors, and computability to other downstream tasks such as inverse problem-solving. Finally, CFG++ achieves remarkable improvements in distillation models (ImageReward of SDXL-Turbo: 0.777 → 0.968, SDXL-Lightning: 0.691 → 0.829). This highlights the unexplored side effects of CFG in low-NFE settings.
>
> **W3.** The statement that the proposed CFG++ is invertible and brings more diversity in the generation is not evaluated.
>
> **A.** We show theoretically why the DDIM inversion error is smaller for CFG++ in Eqs. (21), (22). We empirically validate our argument in Fig. 6, where we show much better reconstruction results. In terms of diversity, it is hard to show that CFG++ enhanced the diversity of the samples. However, as can be seen in the PF-ODE trajectory of the generation shown in the (bottom) of Fig. 1, we can deduce that CFG++ will have a higher chance of being diverse, as the generation is done in a coarse-to-fine manner, as opposed to CFG, where the details are already pre-configured in the earlier stages.
>
> **Typo1,2.** Fixed.
>
> **Typo3.** This should be “renoising”. As can be seen in the difference between Alg. 1 and 2, the crucial difference lies in the difference in line 4, where the noise is added back to achieve.
>
> **Q1.** Average performance across the different corruption in Tab. 3 would help. CFG++ seems more efficient than CFG for FID but not so much according to LPIPS. Any justifications?
>
> **A.** We modified our table to include the average score, demonstrating that our method outperforms others in most aspects. Much of the improvements by using CFG++ instead of CFG were seen in the perceptual quality of the reconstructions, rather than the distortion [2]. FID metrics capture this well, and we think this is the reason why the FID shows the most pronounced improvement among all the different metrics. However, even for LPIPS, we do see that our method outperforms the baselines.
>
>
> **References**
>
> [1] Ho, Jonathan, and Tim Salimans. "Classifier-free diffusion guidance.", 2022.
>
> [2] Blau, Yochai, and Tomer Michaeli. "The perception-distortion tradeoff." CVPR 2018.

---

> > ### Comment · Reviewer_hzx1 · 2024-11-26
> >
> > I would like to thank the authors for their reply. Most of my concerns have been addressed and I am rather positive on the rebuttal.
> >
> > The proposed CFG++ reduces the invertibility error, but this is not equivalent to being invertible as claimed in the paper contributions (l. 107). I suggest that the authors ease this claim in the introduction.
> >
> > In the updated version of the paper, the ImageReward of SD v1.5 (DPM++ 2M) seems particularly low and CFG outperforms CFG++. This result is not commented on. Is there a reason explaining this behavior?

---

> > > ### Author Response · Authors · 2024-11-26
> > >
> > > We would like to thank the reviewer for the constructive feedback. We are glad that most of the concerns were addressed.
> > >
> > > 1. We agree that we could tone the claim down. We modified L107 to "Furthermore, CFG++ reduces the inversion error, enhancing and simplifying image reconstruction, as well as editing"
> > >
> > > 2. This outcome differs from the consistent improvement of CFG++ over CFG observed with 50 NFE DDIM sampling. We attribute this to two factors: (1) the overall image quality for 20 NFE DPM++ 2M is generally lower compared to 50 NFE DDIM, leading to noisier quantitative metrics, particularly for metrics like ImageReward that are sensitive to subtle quality changes; and (2) the difference between CFG and CFG++ tends to be less pronounced in low NFE regimes without distillation, as stronger guidance effects emerge with higher NFEs.
> > > Moreover, the primary goal of this experiment was to demonstrate the compatibility of CFG++ with higher-order solvers, rather than to establish its superiority. The results indicate that CFG++ effectively integrates with such solvers, and its desirable properties become more apparent as the number of NFEs increases.
> > >
> > > We made this clear in the revised **Section 4.1**. Please let us know if you have any further questions.

---

### Author Response · Authors · 2024-11-20
**General Response**

We thank the reviewers for their constructive, positive, and thorough reviews. We are happy that the reviewers think that our paper is **well-written** and **easy to understand** (```hzx1```, ```7gyf```, ```GmUK```, ```NrB8```), **well-grounded with theory** (```GmUK```, ```NrB8```), and conducts **thorough experiments** (```hzx1```, ```GmUK```, ```NrB8```). For a point-to-point response on the weaknesses and the questions, please see our responses below.

---

### Comment · Area_Chair_a9GC · 2024-11-25
**Please check the authors' responses**

Dear reviewers,

Could you please check the authors' responses, and post your message for discussion or changed scores?

best,

AC

---

### Comment · Area_Chair_a9GC · 2024-11-25
**Please focus on the evaluation of the submission**

Dear reviewers and authors,

Let us refocus on the evaluation of the submission itself. The ACs will make a fair decision based on the provided comments and any remaining concerns following the rebuttal and discussion phases.

Reviewer NrB8: Could you please read the authors' responses and the submitted paper to determine if your concerns have been adequately addressed? Additionally, please participate in the discussion and indicate if you have any remaining concerns. In principle, each reviewer should provide sufficient explanations and evidence to justify their rating and confidence level.

Reviewers 7gyf and hzx1, could you please read the authors' responses and post your messages on your opinions?

Best,

AC

---

### Meta-Review · Area_Chair_a9GC · 2024-12-23

**Metareview:**

This paper works on the Classifier-free guidance (CFG) in diffusion models for text-guided generation. It tackles the the off-manifold problem in CFG by proposing a simple revision to the original CFG, resulting in improvements with better sample quality for text-to-image generation. Experimental results demonstrate the effectiveness in text-to-image generation, DDIM inversion, editing, and image inverse problems. Four reviewers (hzx1, 7gyf, GmUK,NrB8) gave scores of 6, 6, 8, 1 after rebuttal. The first three reviewers are positive on the simplicity and experimental justification of improvement over CFG. There are some debates on the fourth reviewer' comments in the post-rebuttal phase. The reviewer NrB8 initially raised several questions and the authors answered these questions in the rebuttal. There is controversy on the reviewer NrB8 's question in the discussion phase. After discussion, the reviewer NrB8 rated score of 1 in the final decision. The AC has asked reviewer NrB8 to provide reasons of decision and remaining concerns on  the authors' responses, but he/she did not respond to the authors' rebuttal. Therefore,  the rating of reviewer NrB8 is not fully supportive. Considering the first three reviewers' positive final decisions, the paper can be accepted, but the authors are suggested to further revise the paper considering these discussions.

**Additional Comments On Reviewer Discussion:**

Reviewer hzx1 questioned on the CFG method's formulation, the claim that CFG++ is invertible and brings more diversity, and also suggested reporting average performance, etc.  Reviewer 7gyf raised questions on comparison with methods like [1], revision on a claim in abstract, validity of CFG/CFG++ with guidance_scale=0.0, etc. Reviewer GmUK is more positive and suggested, e.g.,  working on flow matching-based generative models, extension of CFG++ to other diffusion solvers, etc. These reviewers are mostly satisfied with the authors' rebuttal in the discussion phase. Reviewer NrB8 initially raised several concerns/suggestions on limited scope of experiments, detailed comparison with a broader array of state-of-the-art techniques in diffusion models,  providing a rigorous mathematical framework, how CFG++ scales with increasing model size or complexity, etc. These comments seem to be general, and the authors have answered these concerns and some of these answers have been in the submitted manuscript. There are some debates on the Reviewer NrB8 's comments, but the reviewer NrB8 did not provide supports for his final rating with score of 1. Therefore, the first three reviewers' comments are taken into account with higher weights in the final decision.

---

### Decision · Program_Chairs · 2025-01-22

Accept (Poster)